# The Asgard archaeal ESCRT-III system forms helical filaments and remodels eukaryotic-like membranes

Nataly Melnikov [ID] [1,2], Benedikt Junglas [ID] [3,7], Gal Halbi [4,7], Dikla Nachmias [ID] [1,2], Erez Zerbib [ID] [1,2], Noam Gueta [ID] [1,2], Alexander Upcher [ID] [5], Ran Zalk [ID] [5], Carsten Sachse [ID] [3,6 ✉], Anne Bernheim-Groswasser [ID] [4,5] & Natalie Elia [ID] [1,2 ✉]

## Abstract

The ESCRT machinery mediates membrane remodeling in numerous processes in cells including cell division and nuclear membrane reformation. The identification of ESCRT homologs in Asgard archaea, currently considered the closest prokaryotic relative of eukaryotes, implies a role for ESCRTs in the membrane remodeling processes that occurred during eukaryogenesis. Yet, the function of these distant ESCRT homologs is mostly unresolved. Here we show that Asgard ESCRT-III proteins of the Lokiarcheota self-assemble into helical filaments, a hallmark of the ESCRT system. We determined the cryo-EM structure of the filaments at 3.6 Å resolution and found that they share features of bacterial and eukaryotic ESCRT-III assemblies. Markedly, Asgard ESCRT-III filaments bound and deformed eukaryotic-like membrane vesicles. Oligonucleotides facilitated the assembly of ESCRT-III filaments and tuned the extent of membrane remodeling. The ability of Asgard archaeal ESCRTs to remodel eukaryotic-like membranes, which are fundamentally different from archaeal membranes, and the structural properties of these proteins places them at the junction between prokaryotes and eukaryotes.

**Keywords** Membrane Remodeling; Membrane Repair; Membrane Fission; Protein Polymerization; ESCRT Phylogeny
**Subject Category** Evolution & Ecology

## Introduction

The Asgard superphyla, manifest a unique branch that, according to current evolutionary hypothesis, is considered to be the closest prokaryotic relative of eukaryotes. All Asgard archaeal species were found to encode for Eukaryotic Signature Proteins (ESPs), that carry cellular functions, which until recently were considered as exclusive to eukaryotes (Eme et al, 2023; Liu et al, 2021b; MacLeod et al, 2019;

Spang et al, 2015; Zaremba-Niedzwiedzka et al, 2017). Thus, Asgard proteins potentially hold the core functions of ancient eukaryotic cellular machines. Of particular interest, are ESPs with membrane remodeling capabilities, as these proteins may contribute to the establishment of cellular compartmentalization during eukaryogenesis. Indeed, Asgard archaea encode for several eukaryotic membrane remodeling machineries including the SNARE complex, BAR domains proteins and the ESCRT machinery (Neveu et al, 2020; Spang et al, 2015; Zaremba-Niedzwiedzka et al, 2017). Among these machineries, ESCRTs are the most conserved and are encoded by all Asgard species discovered up to date, suggesting that they carry essential functions in these unique prokaryotes (Hatano et al, 2022; Lu et al, 2020).

The ESCRT machinery constitute one of the most robust and versatile cellular apparatus for membrane constriction and fission. Proteins of the machinery are encoded by all domains of life and were shown to mediate membrane remodeling in a wide range of eukaryotic cellular membranes including plasma membrane, nuclear membrane, and endocytic membranes (Caspi and Dekker, 2018; Hurley, 2015). Moreover, ESCRT-mediated processes span a large spectrum of length scales, from less than 100 nm for vesicle release to up to 1 μm for cell division. How the ESCRT complex mediate membrane remodeling in this large landscape of membrane sources and length scales is mostly unknown (Alonso et al, 2016).

The eukaryotic ESCRT system is composed of five sub-complexes, i.e., ESCRT 0-III and the AAA-ATPase VPS4. Within this complex, ESCRT-III (named CHMPs in animal cells) and VPS4 manifest the minimal unit required for driving membrane fission (Alonso et al, 2016; McCullough et al, 2013). According to current models, membrane recruitment is mediated by early ESCRTs (ESCRT 0-II), while membrane constriction and scission are mediated by polymerization of ESCRT-III proteins into helical hetero-filaments and their remodeling by VPS4, seen in vitro and in cells (Caillat et al, 2019; Goliand et al, 2018; Guizetti et al, 2011; Henne et al, 2012; Lata et al, 2008; Maity et al, 2019; McCullough et al, 2015; McCullough et al, 2013; Mierzwa et al, 2017; Pfitzner et al, 2020). However, the large number of proteins within the ESCRT-III subfamily—twelve in humans and eight in yeast—have challenged mechanistic studies of the basic function of the machine.

[1]Department of Life Sciences, Ben-Gurion University of the Negev, Beer Sheva 84105, Israel. [2]National Institute for Biotechnology in the Negev (NIBN), Ben-Gurion University of the Negev, Beer Sheva 84105, Israel. [3]Ernst-Ruska Centre for Microscopy and Spectroscopy with Electrons, ER-C-3/Structural Biology, Forschungszentrum Jülich, 52425 Jülich, Germany. [4]Department of Chemical Engineering, Ben-Gurion University of the Negev, Beer Sheva 84105, Israel. [5]Ilse Katz Institute for Nanoscale Science and Technology, Ben Gurion University of the Negev, Beer Sheva 84105, Israel. [6]Department of Biology, Heinrich Heine University, Universitätsstr. 1, 40225 Düsseldorf, Germany. [7]These authors contributed equally: Benedikt Junglas, Gal Halbi. ✉E-mail: c.sachse@fz-juelich.de; elianat@post.bgu.ac.il

Reduced, simplified ESCRT-III systems have been identified in prokaryotes including in bacteria and archaea (named Vipp1 and PspA in bacteria and CdvB in archaea) (Caspi and Dekker, 2018; Frohn et al, 2022; Liu et al, 2021a). While the bacterial homologs were shown to remodel membranes, no VPS4 was identified in this system so far. ESCRTs encoded by TACK archaea were shown to participate in cell division, suggesting functional similarities between the eukaryotic and archaeal ESCRT systems (Lindas et al, 2008; Samson et al, 2008; Samson et al, 2011; Tarrason Risa et al, 2020). Within the archaeal domain the Asgard ESCRT system is considerably closer to the eukaryotic system. While all other archaea encode only for ESCRT-III and VPS4 homologs, Asgard archaea encode for homologs of the complete ESCRT system (ESCRT-I-III and VPS4) with Asgard ESCRTs-I and -II exhibiting functional similarities to their eukaryotic homologs including ubiquitin binding (Caspi and Dekker, 2018; Hatano et al, 2022; Lu et al, 2024; Makarova et al, 2024). In addition, the sequence of Asgard ESCRT-III/VPS4 proteins is more closely related to those of eukaryotes than to other archaeal ESCRT systems (named CDV) (Frohn et al, 2022; Lu et al, 2020). Lastly, VPS4 homologs of Asgard archaea were shown to functionally interact with eukaryotic ESCRTs in both yeast and mammalian cells (Lu et al, 2020; Nachmias et al, 2023). Notably, the Asgard ESCRT-III subfamily is substantially reduced compared to eukaryotes with only two ESCRT-III proteins (named here CHMP1-3 and CHMP4-7) encoded in some species including Loki and Heimdall archaeota (Caspi and Dekker, 2018; Lu et al, 2020; Nachmias et al, 2023; Zaremba-Niedzwiedzka et al, 2017). Hence, the Asgard ESCRT system may constitute a more basic version of the ESCRT machinery, which holds the core capabilities of eukaryotic ESCRTs, and could have been involved in membrane remodeling processes that occurred during eukaryogenesis.

Recent cultivation efforts have yielded the isolation of two species of the Asgard Loki phylum (Imachi et al, 2020; Rodrigues-Oliveira et al, 2023). Interestingly, both species are characterized with elongated membrane protrusions and membrane blebs, which are aligned with typical ESCRT functions. Yet, molecular tools for cell biology studies in these species are currently unavailable, precluding analysis of ESCRT function in these cells. Therefore, the function, organization, and properties of Asgard ESCRT-III complexes as well as their ability to remodel membranes are currently unknown.

In this work, we provide the first evidence that Asgard ESCRT-III complexes share functional similarities with eukaryotic ESCRTs. Using purified ESCRT-III homologs encoded by the most abundant Asgard phyla, Lokiarchaeota (Loki), we show that the Asgard ESCRT-III protein CHMP4-7 can self-assemble, on its own and in the presence of CHMP1-3, into helical tubes that resemble those of eukaryotes (Azad et al, 2023; Henne et al, 2012; Lata et al, 2008; McCullough et al, 2015). The frequency of helical tubes was increased in the presence of short ssDNA oligos, while no filaments could be observed upon DNAse treatment. Next, we determined the cryo-EM structure of helical CHMP4-7 tubes in the presence of CHMP1–3 and DNA at a resolution of 3.6 Å. We found that the 36 nm wide tubular assemblies are built from protomers adopting the canonical ESCRT-III-fold that share characteristic intersubunit contacts with both bacterial and eukaryotic ESCRT-III polymers. Finally, we found that Loki ESCRT filaments, formed under any of these conditions, are able to bind and deform small unilamellar vesicles (SUVs) comprised of eukaryotic-like synthetic phospholipids. Hence, Loki ESCRT-III assemble into helical filaments that are affected by oligonucleotides and are capable of remodeling eukaryotic-like membranes.

# Results

Current phylogenic models suggest that eukaryotes have branched from a lineage closely related to the archaeal Asgard clade (Fig. 1A) (Eme et al, 2023). ESCRT-III proteins encoded in eukaryotes polymerize into helical filaments—a hallmark of ESCRT-III organization that is thought to be essential for their function in membrane remodeling (Hurley, 2015; Olmos, 2022). To investigate the polymerization properties of Asgard ESCRT-III proteins, we purified recombinant versions of the ESCRT-III proteins CHMP4-7 and CHMP1-3, encoded by the Loki GC14_75 strain (KKK44605.1 and KKK42122.1, respectively). Purified proteins were then subjected to polymerization reaction (see "Methods") and visualized by negative-stain TEM and cryo-EM. Recombinant full-length CHMP4-7 spontaneously self-assembled into helical filaments (Fig. 1B), while no evidence for polymerization could be detected for purified CHMP1-3 using a range of experimental conditions (Appendix Fig. S1). The overall shape of CHMP4-7 filaments, seen by negative-stain EM, resembled those previously described for eukaryotic or bacterial ESCRT-III filaments (Henne et al, 2012; Junglas et al, 2021; Lata et al, 2008; McCullough et al, 2015). CHMP4-7 polymers were predominantly seen as long, thin and curved filaments or as rod-shaped tubes (Fig. 1B, see arrows and Fig. 1C). A Moiré pattern of parallel, periodic stripes was detected in the CHMP4-7 tube structures by cryo-EM, supporting a helical tube organization (Fig. 1C,D). The outer diameter of the CHMP4-7 tubes was variable, ranging between 35 and 55 nm (averaged diameter 44 ± 30.9 nm). Tube length was typically between 300 and 450 nm (Fig. 1E). The diameter of Loki CHMP4-7 helical tubes was slightly larger than the diameters measured for human CHMP1B (~25 nm) or the bacterial ESCRT-III homologs PspA (~21 nm), but was in the range of diameters measured for human CHMP2A-CHMP3 (38–43 nm) (Junglas et al, 2021; Lata et al, 2008; McCullough et al, 2015). Collectively, these data indicate that Loki CHMP4-7 spontaneously self-assemble into the typical helical tube organization described for the ESCRT-III complex.

Next, we examined the effect of CHMP1-3, which did not polymerize on its own, on Loki ESCRT-III filaments. To this end, full-length Loki CHMP4-7 and CHMP1-3 were co-incubated at different ratios and subjected to polymerization reaction. While no polymers were seen at 1:1 CHMP4-7-CHMP1-3 ratio, helical tubes could be readily observed at 2:1, 4:1, and 6:1 ratios (Fig. 2; Appendix Fig. S2A). Polymerization was completely abolished upon using a C terminally truncated CHMP1-3 mutant (AAs 1-163) at 4:1 ratio, supporting a role for the C' of CHMP1-3 in CHMP4-7-CHMP1-3 filament formation (Appendix Fig. S2B). Helical tubes assembled in the presence of full-length CHMP4-7 and CHMP1-3 were morphologically distinct from helical tubes assembled using only CHMP4-7. First, their outer diameter was significantly smaller and less diverse (28 ± 1.6 nm vs 44 ± 30.9 nm for CHMP4-7/CHMP1-3 4:1 ratio and CHMP4-7 alone, respectively) (Fig. 2B, left panel). Second, helical tubes formed in the presence of both CHMP4-7 and CHMP1-3 were significantly longer than those formed by the CHMP4-7 homopolymer (> X1.5 folds increase for 2:1 and 4:1 ratios) (Fig. 2B, right

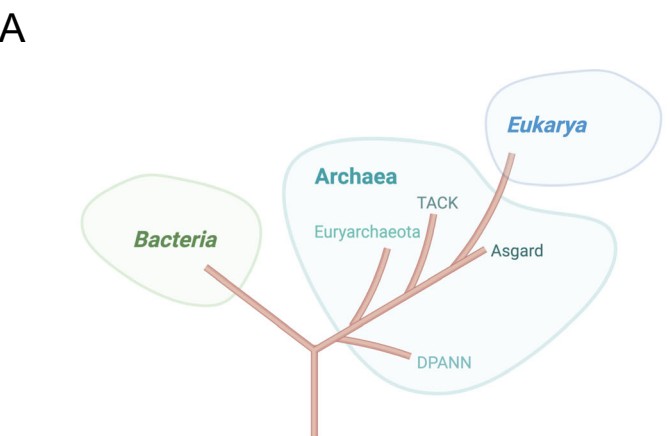

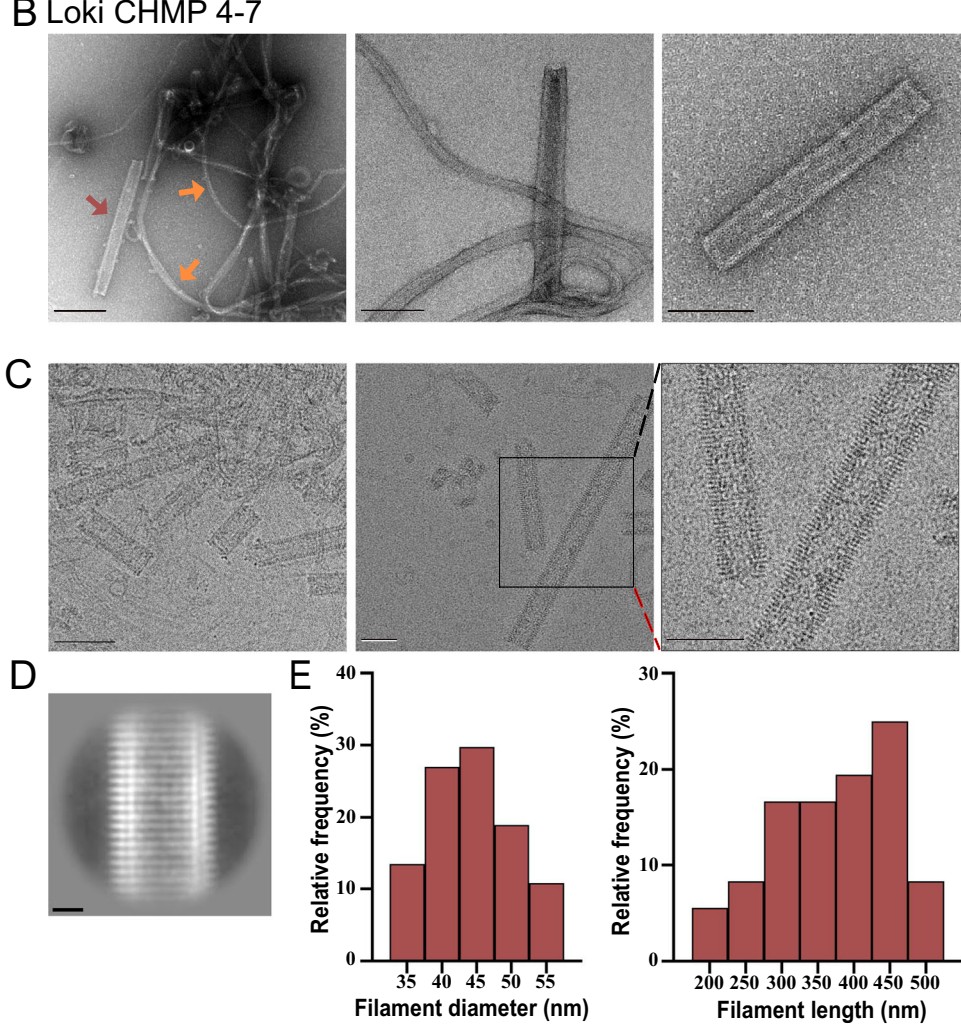

**Figure 1. Loki CHMP4-7 self-assembles into helical tubes.**

(A) Schematic representation of the currently suggested models for the evolutionary relationships between Bacteria, Archaea, and Eukarya that integrate the archaeal Asgard clade. (B) Negative-stain TEM micrographs of Loki CHMP4-7 homopolymers. Representative zoomed-out (left, scale = 200 nm) and zoomed-in (middle and right, scale = 100 nm) images are shown. Orange arrows indicate proto-filaments, red arrow indicates a helical tube. (C) Representative Cryo-EM micrographs of Loki CHMP4-7 homopolymers. Left, scale = 100 nm, middle and right, scale = 50 nm. An enlargement of the area depicted in black box in middle panel is show in right panel. (D) A representative 2D class average of 502 particles extracted from CHMP4-7 projections. Scale = 100 Å. (E) Frequency histograms of Loki CHMP4-7 helical diameters (left, $n = 37$) and length (right, $n = 36$) obtained from negative-stain TEM images acquired from two independent experiments. Averaged CHMP4-7 helical tube diameter 44 ± 30.9 nm. Source data are available online for this figure.

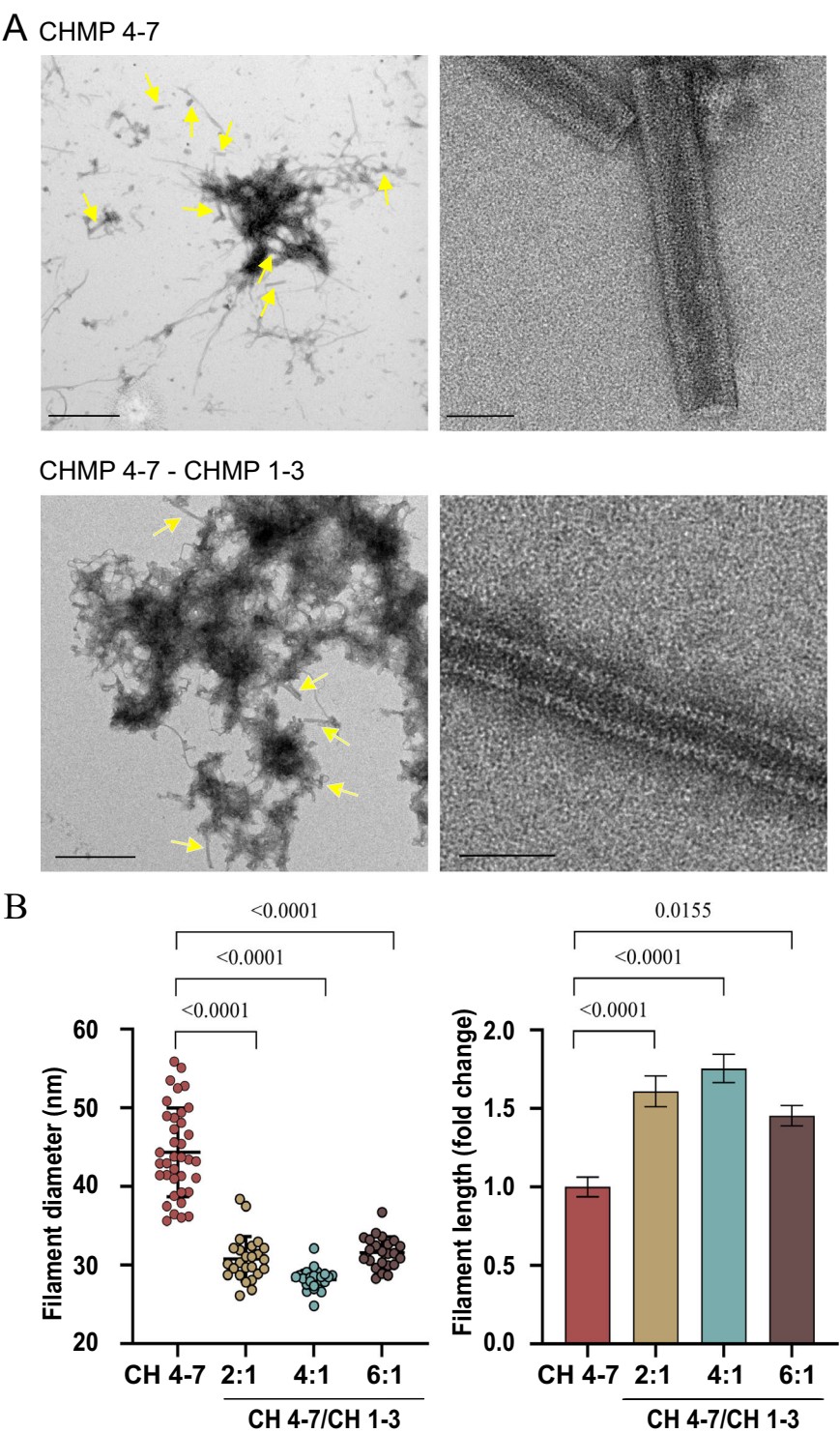

**Figure 2.   CHMP1-3 induces morphological changes in Loki ESCRT-III helical tubes.**

(**A**) Representative negative-stain TEM images of Loki CHMP4-7 homopolymers (upper panels) and CHMP4-7-CHMP1-3 co-polymers (bottom panels, 4:1 molar ratio). Zoomed-out (left, scale = 1 μm) and zoomed-in (right, scale = 50 nm) images are shown. Yellow arrows indicate helical tubes. Note, that the helical tube in co-polymers is more curved and displays periodic stripes that are not seen in the homopolymers by negative staining. (**B**) Diameter (left) and length (right) measurements of CHMP4-7-CHMP1-3 helical tubes that were assembled at the indicated molar ratios. Measurements were obtained from negative-stain TEM images. Averaged diameters (nm): 2:1 ratio, $31 \pm 7.7$, $n = 25$; 4:1 ratio $28 \pm 1.6$, $n = 26$, 6:1 ratio $32 \pm 3.8$, $n = 23$. Averaged lengths (nm): 2:1 ratio $641 \pm 173$, $n = 25$; 4:1 ratio $594 \pm 158$, $n = 27$; 6:1 ratio $492 \pm 109$, $n = 25$. $n$ refers to number of filaments. Measurements were obtained from at least two independent experiments for each condition. Statistical analysis, one-way analysis of variance (ANOVA) Kruskal–Wallis test followed by Dunn's post hoc test was performed (see "Methods"). Error bars: left panel, STDEV, right panel, standard error. Source data are available online for this figure.

panel). Third, the morphology of the CHMP4-7/CHMP1-3 helical tube was slightly different: tubes appeared to be more curved and periodic stripes, were readily detected in negative-stain images, suggesting that packing of the helical tube is less tight in the presence of CHMP1-3 (Fig. 2A). In addition, unpacked filaments, resembling a telephone cord were observed, further supporting a continuous helical tube assembly (Appendix Fig. S2C). The phenotypic changes observed in the presence of CHMP1-3 were not induced upon substituting CHMP1-3 with purified GFP or by increasing the concentration of CHMP4-7, suggesting that they are specifically contributed by CHMP1-3 (Appendix Fig. S2D). Therefore, the Loki ESCRT-III protein CHMP4-7 self-assemble into helical filaments, with CHMP1-3 affecting the properties of these filaments. These findings are in line with previous observations obtained for human ESCRT-III CHMP1B helical filaments, which also formed a narrower, more uniform tubes in the presence of IST1 (Nguyen et al, 2020). The realization that both human and Asgard ESCRT-III proteins assemble into helical filaments that can adopt a narrower, more uniform tube shape in the presence an additional subunit, suggests that this is a conserved property of the ESCRT-III system.

Recent work from our laboratory showed that purified Loki CHMP4-7 binds short DNA oligonucleotides (Nachmias et al, 2023). Human CHMP1B, exhibited similar properties and a recently solved cryo-EM structure of human CHMP1B/IST1 filaments demonstrated direct interactions between the ESCRT-III filament and nucleic acids (Nachmias et al, 2023; Talledge et al, 2018). We, therefore, set to examine the effect of short DNA oligonucleotides on the self-assembly properties of Asgard ESCRT-III filaments. A five-fold increase in the occurrence of helical tubes was observed for CHMP4-7/CHMP1-3 helical tubes that were assembled in the presence of 40 bases long ssDNA, in negative-stain TEM images (Fig. 3A–C; Appendix Fig. S3A). While Loki CHMP4-7 was able to bind both the ss and dsDNA forms comprising the same sequence, using dsDNA did not enhance helical tube formation (Appendix Fig. S3B,C) (Nachmias et al, 2023). Incubating CHMP1-3 or the CHMP1-3 C' deletion mutant with ssDNA, did not induce polymerization, further supporting the inability of CHMP1-3 to self-assemble in the absence of CHMP4-7 (Appendix Fig. S3D). The presence of DNA affected the properties of CHMP4-7/CHMP1-3 helical tubes. Helical tubes formed in the presence of ssDNA were wider and slightly longer (averaged diameters: $40 \pm 4$ nm vs. $29.4 \pm 4.6$ nm; averaged lengths: $777.5 \pm 210$ nm vs. $535 \pm 119$ nm for helical tubes polymerized with and without ssDNA, respectively) (Fig. 3D). Moreover, filaments were less curved, and the periodic stripes observed in negative-stain images could not be detected (Fig. 3A). Addition of ssDNA also increased the frequency of helical tubes formed by CHMP4-7 alone, but to a lesser extent (three folds), without affecting their characteristic length or diameter (Fig. 3B–D; Appendix Fig. S3E). Analysis of different ssDNA sequences revealed that oligos that are 40–80 bases long and comprise GC nucleotides induce these effects (Appendix Fig. S4). Helical tube formation was completely abolished for both CHMP4-7/CHMP1-3 and CHMP4-7 filaments in the presence of DNase, resulting aggregates formation, further supporting a role for DNA in ESCRT-III helical tube assembly, in vitro (Fig. 3E; Appendix Fig. S3F).

Given the higher frequency of ESCRT-III helical filaments obtained for CHMP4-7/CHMP1-3 filaments in the presence of ssDNA, we set out to resolve the cryo-EM structure of the filaments. In our cryo-EM micrographs, we found two types of filaments: straight tube/rod-like polymers with variable diameters

and thinner highly curved polymers (as described above) (Fig. 4A; Appendix Fig. S5A). First, the curved filaments had diameters of 6–15 nm and showed distinct features in their 2D class averages such as saw-tooth-like spikes emanating sideways from the filament including, in some cases, a central smooth line (Appendix Fig. S5B). Nevertheless, we were unable to generate reliable 3D reconstructions, likely due to their variable curvature and remaining heterogeneity. Secondly, the rod-shaped filaments had diameters ranging from 30 to over 45 nm. The 2D class averages showed well-defined spikes at the filament rims and a Christmas tree like pattern (Fig. 4B) overall, closely resembling PspA rods (Junglas et al, 2021). The power spectra of 2D classes of increasing diameters revealed highly similar layer line patterns, indicating a very similar helical architecture of the rods with increasing number of units per turn (Appendix Fig. S5C). We were able to determine the structure of 36 nm PspA-like rods at a global resolution of 3.6 Å and a local resolution ranging from 3.0 Å at the inner rod wall to 6.0 Å at the outer tube wall (Fig. 4C–E). The rods are arranged by helical symmetry with 1.05 Å rise and 173.9° rotation. In the lumen of the rods, weak and fuzzy cylindrical density was found close to the inner wall of the rods at ~13 nm radial position (Appendix Fig. S5D,E). The density is continuous and does not follow the imposed helical symmetry while it gives rise to a weak peak in the radial density profile distanced 32 Å away from the main protein peak (Appendix Fig. S5E). Thus we, relied on the strong density to build a model for Loki CHMP4-7 α1–5 into the density (Fig. 5A,B), while no density was found for helices α0 and α6. Although we could not clearly assign any density corresponding to CHMP1-3 and ssDNA in the cryo-EM map, biochemical analysis using specific antibodies showed that they reside in the filament fraction, suggesting that they are associated with the filament (Appendix Fig. S6). CHMP1-3 is, therefore, not a regular member of the symmetric rod assemblies while it may be part of the curved filaments, involved in nucleation or occasionally bound to the rod assemblies. The structured elements of CHMP4-7 adopt the canonical ESCRT-III fold: α1–3 form an extended hairpin, connected to α4 via a short linker (E117-P120 Fig. 5B, inset). Helix α4 is separated in two parts by a short loop at G146 and connected to α5 by an extended linker (I154-E159). The monomers are arranged in a $j + 4$ repeat, where α5 of chain j packs perpendicularly against the tip of the hairpin of $j + 4$ (Fig. 5C), forming the typical ESCRT-III assembly motif known from IST1/CHMP1B (McCullough et al, 2015). In the assembly, helices α1-4 make up the wall of the rods, while α5 together with the tip of the $α1 + 2$ hairpin forms the spikes on the outside of the wall (Appendix Fig. S5F,D). Consequently, although unresolved, α6 is likely located outside the rods while α0 points into the lumen perhaps as part of the weak density found inside the rods. Noteworthy, α0 is positively charged and could possibly interact with the lumen of the rods to form the observed cylindrical density. The electrostatic surface of the rod assembly reveals a distinct polarity with one negatively and one positively charged end (Appendix Fig. S5G). The same interfaces likely also stabilize the layers of the rod assembly by electrostatic interactions.

Loki CHMP4-7 adopts the canonical ESCRT-III fold in the open form and the assembly resembles typical features of ESCRT-III tubes including the conserved motif between α1/2-α5. Comparing the CHMP4-7 rods with ESCRT-III tube-like assemblies of other domains of life, we found that their architecture is most similar to the bacterial assemblies that have a α5-hairpin structure emanating

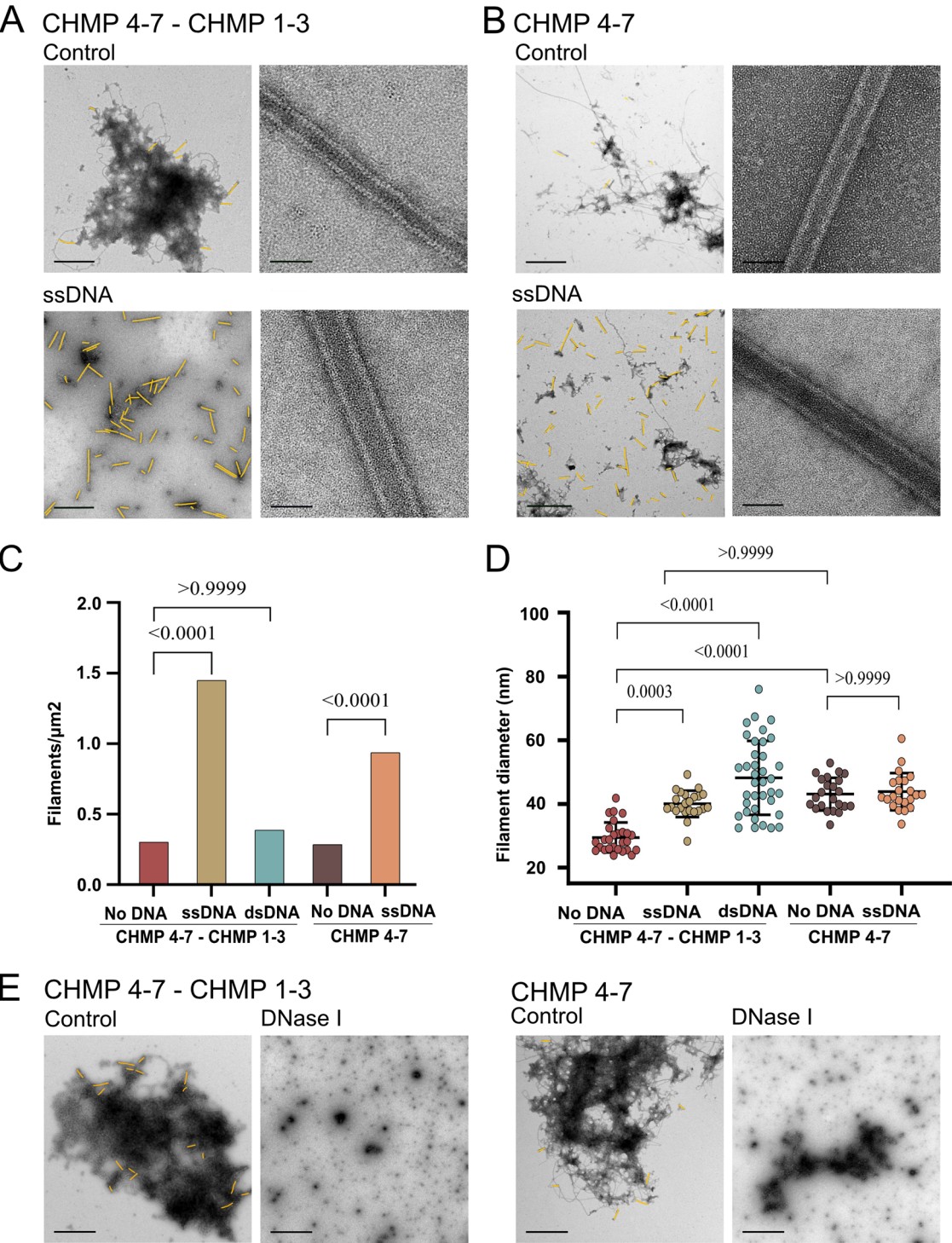

**Figure 3. DNA facilitates Loki ESCRT-III helical tube assembly.**

(A–D) Loki ESCRT-III proteins CHMP4-7 and CHMP1-3 (4:1 molar ratio) (A) or CHMP4-7 alone (B) were subjected to polymerization reaction (see "Methods") in the absence or presence of 40 bases oligonucleotides (ssDNA or dsDNA, as indicated), and documented using negative-stain TEM. Representative zoomed-out (left, Scale = 1 μm) and zoomed-in (right, scale = 50 nm) images are shown in (A, B) and in Appendix Fig. S3A,B. Helical tubes are colored in yellow in zoomed-out images (raw images, Appendix Fig. S3A). (C) Helical tubes density at various conditions. Measurements were performed on an area >565 μm$^2$ from each experiment. Data from two independent experiments are presented for each condition (using at least 30 images from each repetition). (D) Diameters of CHMP4-7 and CHMP4-7/CHMP1-3 helical tubes assembled in the presence of oligonucleotides. Averaged diameters (nm): CHMP4-7/CHMP1-3 29.4 ± 4.6, n = 27; CHMP4-7/CHMP1-3 ssDNA 40 ± 4.1, n = 23; CHMP4-7/CHMP1-3 dsDNA 48 ± 11.5, n = 37; CHMP4-7 43 ± 5, n = 23; CHMP4-7 ssDNA 44 ± 5.7, n = 22. n refers to number of filaments. Measurements were obtained from at least two independent experiments for each condition. Error bar, STDEV. (E) Loki ESCRT-III proteins CHMP4-7 and CHMP1-3 (4:1 molar ratio) (left) or CHMP4-7 alone (right) were assembled as above, or in the presence of 0.02 mg/ml DNase, as indicated. Helical tubes are colored in yellow (raw images, Appendix Fig. S3F). Scale = 1 μm. Data were reproduced in four independent experiments. Statistics in (C, D) was performed using one-way analysis of variance (ANOVA) (see "Methods"). Source data are available online for this figure.

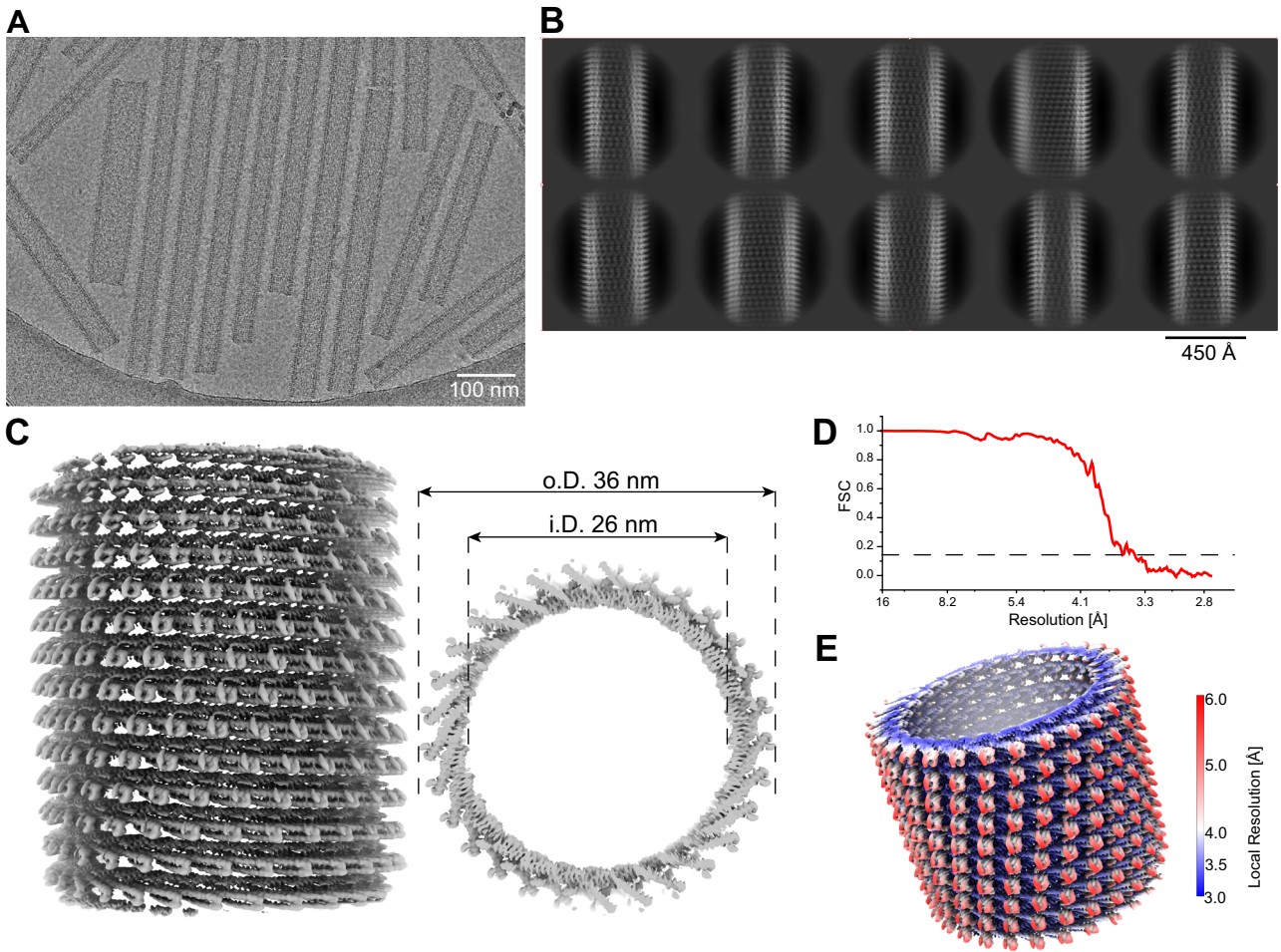

**Figure 4. Loki CHMP4-7 forms PspA-like rod structures.**

(A) Example cryo-EM micrograph of Loki CHMP4-7 showing straight rod structures with different diameters. (B) Set of 2D class averages of Loki CHMP4-7 rods with the typical Christmas tree pattern of PspA rods. (C) Side (left) and top (right) view of cryo-EM density map of the rod structure with an inner and diameter of 26 and 36 nm, respectively. (D) Fourier shell correlation of the cryo-EM density map indicating a nominal resolution at 3.6 Å according to the FSC 0.143 cutoff. (E) Estimates of local resolution mapped in color code (blue = 3.0 and red 6.0 Å resolution, respectively) on the cryo-EM density. Source data are available online for this figure.

from the tubular wall and giving rise to the characteristic spikes in the projections (Fig. 5E, top). In contrast, in eukaryotic assemblies such as IST1 and CHMP1B the α5-hairpin structure does not protrude from the assembly, thereby forming smooth tubes. In CHMP4-7 rods of Asgard, α5 only interacts with a single hairpin at the bottom end, which is a common feature in eukaryotic ESCRT-III assemblies, while in bacteria top and bottom ends of α5 are interacting with two hairpins connecting two layers of the assembly (Fig. 5E, bottom). In addition, on the monomer level, CHMP4-7 more closely resembles the eukaryotic ESCRT-III proteins: first, the bacterial ESCRT-IIIs have more extended α0 and α2 + 3 helices (corresponding to aa 10–20, aa 108–123 in the consensus sequence Appendix Fig. S5H). Secondly, in the bacterial ESCRT-IIIs the linker between α3 and α4 is longer and contains more helix-breaking residues (i.e., G156, G157, and G159 in Vipp1). Therefore, the Loki CHMP4-7 assembly more closely resembles the bacterial configuration with the characteristic spike, while the Loki CHMP4-7 monomer adopts a conformation closer to previously determined eukaryotic structures including the α1/2-α5 interaction motif.

Eukaryotic and bacterial ESCRT-IIIs were shown to bind and deform membranes in vitro and in cells (Gupta et al, 2021; Junglas et al, 2021; Liu et al, 2021a; Olmos, 2022). To determine whether ESCRTs encoded by Asgard archaea carry the ability to remodel membranes, we incubated pre-assembled Loki ESCRT-III filaments with small unilamellar vesicles (SUVs, ~100 nm) composed of negatively charged eukaryotic-like phospholipids (1:1, PC:PS). Under these conditions, ESCRT-III filaments, were found to intimately interact with SUVs under any of the conditions tested (including in the presence of ssDNA) (Fig. 6A,B; Appendix Fig. S7). We identified four main types of ESCRT-III-SUV interactions: (1) Attachment of the SUV to the outside surface of the tube, which was associated with flattening of the vesicle at the filament-vesicle interface; (2) Docking of the SUV at the tip of the helical tube; (3) Invagination of the SUV to the interior of the ESCRT-III tube; and (4) Dramatic remodeling of both the SUV and the ESCRT-III tube (see examples in Fig. 6C, left panel). Combinations of several types of interactions on the same filament were also observed, indicating that these interactions are not mutually exclusive. In addition, we

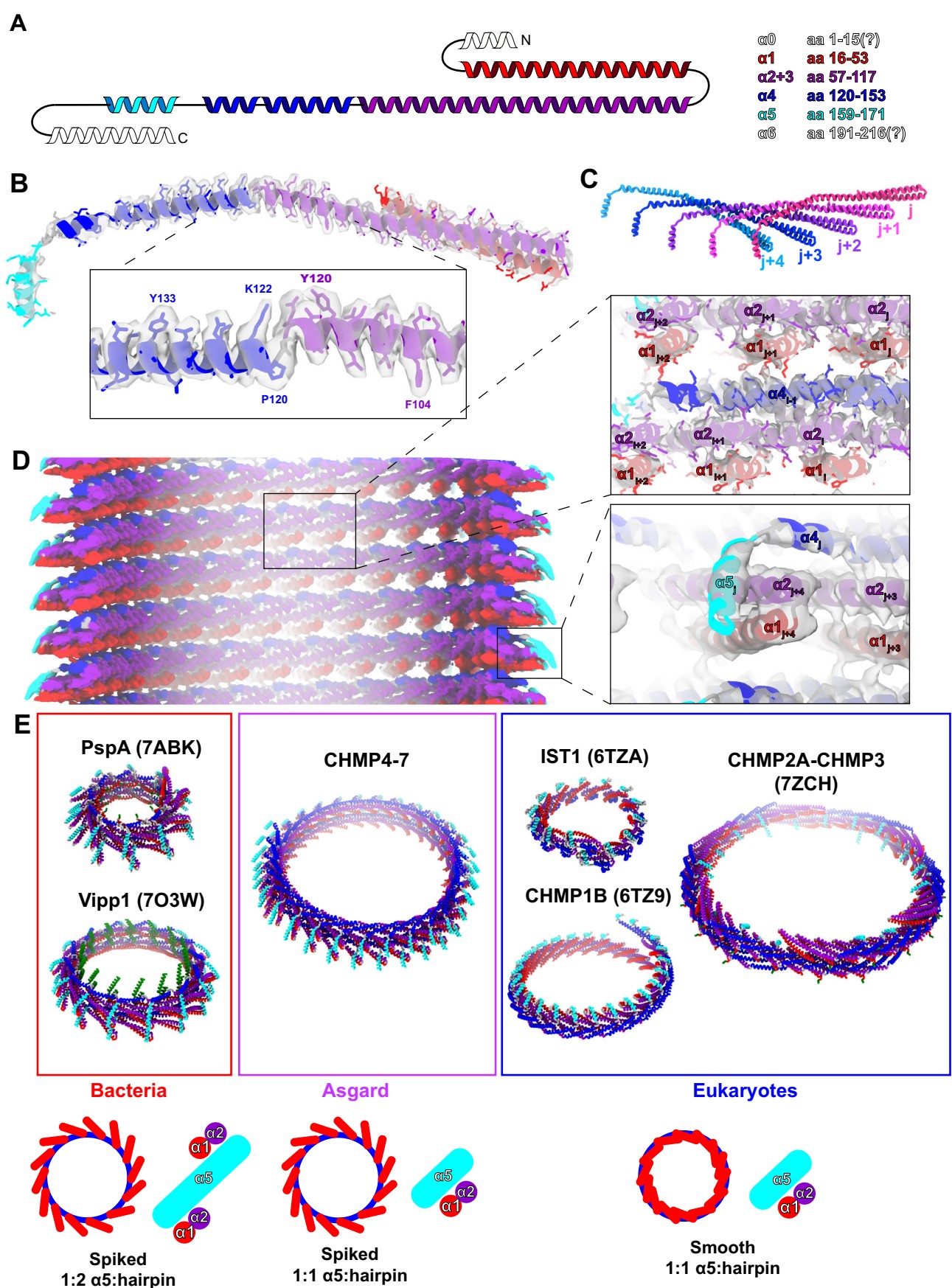

**Figure 5.   Loki CHMP4-7 rods are built from monomers sharing typical ESCRT-III features.**

(A) Topology plot of the Loki CHMP4-7 structure with color-coded helices: α0 white, α1 red, α2 + 3 violet, α4 blue, α5 cyan, α6 white. (B) Segmented density including modeled atomic Loki CHMP4-7 monomer structure (inset visible side-chains aa 104–134) found in 36 nm rods showing the ESCRT-III fold. (C) Organization of the CHMP4-7 monomers in the helical assembly: α5 of chain j interacts with the hairpin of chain j + 4. (D) Cryo-EM map of Loki CHMP4-7 rod with isosurface colored CHMP4-7 α-helices and enlarged view of the cryo-EM map with the fitted model showing the interactions in the tube wall and helix α5 interactions in the periphery. (E) Comparison of tubular ESCRT-III assemblies from bacteria, Asgard archaea and eukaryotes. PspA: 7abk, Vipp1: 7o3w, IST1: 6tza, CHMP1B: 6tz9, CHMP2A-CHMP3: 7zch. Color-coded helices: α0 green, α1 red, α2 + 3 violet, α4 blue, α5 cyan.

occasionally observed a docked SUV invaginating into the helical tube in cryo-EM images and tomograms (Fig. 6B, left panel, D; Movies EV1–3), suggesting that these two types of interactions represent successive steps. Invaginations of more than one vesicle to the same tube could be observed, giving rise to the formation of a complex membrane system with several vesicles internalized into one another (Fig. 6B; Movies EV1–3). The characteristic periodic stripes of the ESCRT-III filament could often be detected at the exterior of the filament-vesicle complex (Fig. 6B,D; Movie 3), suggesting that the observed morphologies result from internalization of vesicles into the ESCRT-III helical tube followed by deformation of the tube. Loki ESCRT-III filaments also interacted with neutrally charged SUVs (100% PC) but to a lesser extent, (27% vs. 15% of the total SUVs interacted with ESCRT-III tubes upon incubation with PC:PS vs PC vesicles, respectively), predominantly exhibiting attachment of the SUV to the exterior of the helical tube (Appendix Fig. S8). Therefore, Loki ESCRT-III filaments associate with membrane vesicles and deform vesicles via the interior of the tube (see proposed model in Fig. 6E).

Last, we asked whether the composition of ESCRT-III helical tubes, described here, affect their capabilities to remodel membranes. To this end, we quantified the frequencies of the different types of filament-vesicle interactions we defined, at the various assembly conditions. No difference in frequencies were observed for helical filaments that assembled in the presence or absence of CHMP1-3 (CHMP4-7/CHMP1-3 vs CHMP4-7), suggesting that CHMP1-3 does not significantly contribute to membrane interactions. The presence of DNA, however, affected the distributions of ESCRT-III-SUV interactions, as follows (Fig. 6C): First, the association of vesicles with the outside surface of the helical tube (outside, type 1) was considerably enriched (No DNA, <15%; with DNA, >35%). Second, interactions with the tip of the filament (docking, type 2) and invagination of the vesicle (inside, type 3) were significantly enriched. Finally, the most significant difference was the abundance of the complete remodeling events (remodeling, type 4) (no DNA ~80%, with DNA, ~5%;), suggesting that the presence of DNA inhibits this terminal tube-vesicle interaction stage. The effect of DNA on Loki ESCRT-III-induced membrane remodeling appeared to be irreversible, as the addition of DNase to pre-established tube-SUV complexes did not lead to extensive membrane remodeling (Appendix Fig. S7C). Therefore, Loki ESCRT-III filaments are capable of interacting and remodeling lipid membranes, both in the presence and absence of ssDNA, but the extent of remodeling appear to be attenuated by the presence of DNA.

## Discussion

In this work, we provide the first evidence to show that, Asgard ESCRT-III proteins assemble into helical tubes that resemble those of their eukaryotic homologs and remodel membranes. These findings confirm that ESCRT homologs encoded by Asgard archaea are functionally related to their eukaryotic homologs. We further provide data to suggest a role for DNA in the assembly of Asgard ESCRT-III filaments. Finally, by using SUVs composed of eukaryotic phospholipids, which are fundamentally different from archaeal lipids, we demonstrate that Asgard ESCRT-IIIs could, in-principle, contribute to the membrane remodeling processes that occurred during eukaryogenesis and gave rise to the complex phospholipids-based endomembrane system of eukaryotes.

Our data point to similarities between the human and Asgard ESCRT-III systems. First, the nature of self-assembly appear to be conserved. Similar to the human CHMP1B/IST1 complex, Loki CHMP4-7 was found to polymerize alone or in the presence of CHMP1-3, with the former exhibiting a wider, more variable diameter (Nguyen et al, 2020). What controls the diameter of the ESCRT-III helical tube is currently unknown. Notably, while IST1 was found to form a layer on the outside of the CHMP1B tube, CHMP1-3 could not be detected in our cryo-EM structure suggesting that it obtains a different mode-of-operation. Regardless of the mechanism, the realization that both human and Asgard ESCRT-III filaments can adopt a wide range of diameters that can be tuned by the addition of another ESCRT-III components, strongly suggest that this is a conserved feature of the ESCRT-III system. Therefore, a basic feature of the ESCRT-III system may be their ability to assemble into adaptable filaments that can be tuned, by additional ESCRT-III proteins, to adopt specific shapes. Such a model can explain how the ESCRT-III system executes the large repertoire of membrane remodeling processes in cells, which greatly vary in scales and topologies (Caspi and Dekker, 2018; Hurley, 2015).

The determined cryo-EM structure of CHMP4-7 helical filaments reveals the canonical ESCRT-III fold made of 5 α-helices including the conserved α1/2-α5 packing equally present in Loki archaea. For eukaryotes and Loki archaea, we detected a number of residue omissions in helices α2 + α3 and the linker between α3 and α4, whose associated properties appear to be dispensable in the context of ESCRT evolution and may be responsible for functional diversification of different isoforms. The basic structural arrangement of the molecule in the CHMP4-7 assembly resembles that of previously determined bacterial ESCRT-III proteins PspA and Vipp1 (Gupta et al, 2021; Junglas et al, 2021). In both cases, the hairpin formed by α1/α2 emanates from the assembly giving rise to characteristic spikes in the EM images. Another feature that is closer to the determined eukaryotic assemblies CHMP1B and IST1 is that helix α5 contacts only a single α1/2 hairpin as opposed to contacting two α1/2 hairpins described for bacterial ESCRT-III proteins (McCullough et al, 2015). This feature makes eukaryotic as well as Loki rods less tightly packed potentially offering additional access for binding partners and thus opening up the possibility to functional modulation and regulation of membrane remodeling by other isoforms of the ESCRT-III family.

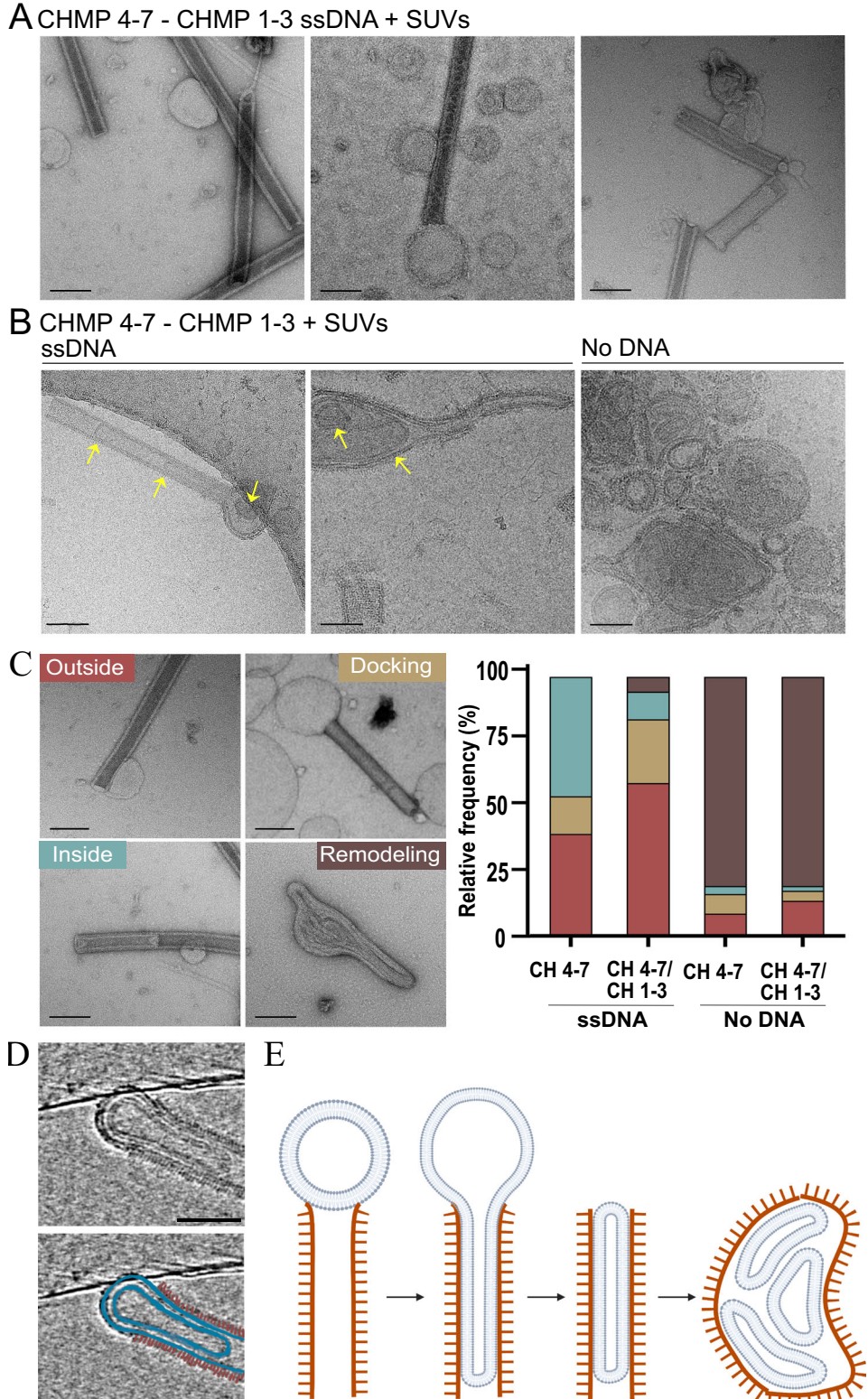

Our data show that CHMP4-7 self-assembles into helical tubes and that addition of CHMP1-3 shifts the observed populations of the Loki ESCRT-III helical tube types, resulting in more uniform and ordered structures suitable for structure determination. How CHMP1-3 induce this effect is currently unknown. While we could not detect CHMP1-3 in the cryo-EM structure, biochemical analysis shows that it resides in the filamentous fraction. Moreover, a CHMP1-3 C' truncation mutant abolished filament formation strongly supporting association between CHMP4-7 and CHMP1-3. Interestingly, we found that the concentration of CHMP1-3 must be lower than that of CHMP4-7 to facilitate tube

**Figure 6.  Loki ESCRT-III filaments deform eukaryotic-like membrane vesicles.**

(A–B) Representative negative-stain TEM (A) and cryo-EM (B) images of CHMP4-7-CHMP1-3 helical tubes (molar ratio, 4:1) assembled in the presence of ssDNA (10 μM), and incubated with SUVs (DOPC:DOPS, 1:1 ratio). Scale = 100 nm. Arrows in (B) specify internalized vesicles. (C) Percentages of the different types of vesicle-tube interactions observed under the specified conditions. Left panel, representative negative-stain images of the four types of interactions quantified. Outside (type 1); Docking (type 2); Inside (type 3); Remodeling (type 4). Data was quantified from negative-stain images obtained from at least two independent experiments for each condition (n = CHMP4-7 ssDNA, 124; CHMP4-7-CHMP1-3 ssDNA, 159; CHMP4-7, 197; CHMP4-7-CHMP1-3, 161). Scale = 100 nm. Pearson chi-squared analysis show a significant difference in the distributions of homo- and co-polymers assembled in the presence vs absence of DNA (P value ≤ 0.0001 for all pairs). No significance was found in the distributions obtained for ESCRT-III filaments assembled in the presence or absence of CHMP1–3 that were subjected to similar treatments. Scale = 1 μm. (D) An overlay of selected sequential slices (50) from a representative reconstructed tomogram of cryo-fixed filaments-SUV complexes. Membranes and ESCRT-III filaments were artificially colored in bottom panel (membrane, blue; filament, brown). The complete series is provided in Movie EV3. Scale = 50 nm. (E) A suggested model for Loki ESCRT-III mediated vesicle remodeling. Vesicles docked at the tip of the ESCRT-III tube directly interact with the interior of the tube, leading to deformation and invagination of the vesicle by the ESCRT-III filament. This process occurs repeatedly, leading to the formation of a complex membrane network inside the helical tube, which ultimately causes deformation and remodeling of the ESCRT-III tube. Tube remodeling is inhibited in the presence of DNA, resulting in the accumulation of ESCRT-III-SUV interactions in all the stages that precede this stage. Source data are available online for this figure.

formation. Collectively, these findings indicate that CHMP1-3 is part of the ESCRT-III helical tube but is not bound regularly throughout the tubular filament in a consistent, recurring fashion to allow its visualization by cryo-EM. One possible scenario for the mechanistic function of CHMP1-3 in the Asgard ESCRT system, is that it binds CHMP4-7 during the nucleation phase and regulates early stages of polymerization. However, more work is needed in order to elucidate the mode-of-operation of CHMP1-3 in the Asgard ESCRT-III system.

The topology of membrane binding, described here for Loki ESCRT-IIIs, involves invagination of the membrane by the ESCRT-III tube. Although the conventional topology attributed to the ESCRT complex is reversed, as described for the CHMP2A-CHMP3 co-polymer (Azad et al, 2023; Lata et al, 2008), membrane binding via the interior of the tube has been reported for the human CHMP1B/IST1 polymer (Nguyen et al, 2020), and for the bacterial ESCRT-III homologs Vipp1 and PspA (Gupta et al, 2021; Junglas et al, 2021). Consistently, in the cryo-EM structure of Loki ESCRT-III the N terminus, which is associated with membrane binding, is facing the interior of the helical tube. Whether internal membrane binding topology represents the preferred topology of ancient ESCRTs or is a consequence of the in vitro experimental system is yet to be determined. Regardless, the similar in vitro phenotypes observed for ESCRTs homologs in prokaryotes and eukaryotes suggests common functional properties for these proteins.

Our results demonstrate that DNA can facilitate Loki ESCRT-III polymerization. Self-assembly of the Loki ESCRT-III polymers into high-ordered helical tubes was enhanced in the presence of short ssDNA oligonucleotides, and helical tubes could not be found in the presence of DNase. Moreover, oligos comprised of sequences lacking GC failed to enhance polymerization supporting sequence-based interactions. That said, we could not assign a density for DNA in the cryo-EM structure, likely due to the lower occupancy and flexible conformations of short oligonucleotides. Possibly, the unresolved weak cylindrical density seen in the lumen of the rods of the Loki ESCRT-III cryo-EM structure represent DNA but it may equally represent parts of CHMP1-3, as both DNA CHMP1-3 were found in the filament fraction by biochemical assays. Notably, ESCRT-Nucleotide interactions were previously reported for other ESCRT systems. First, nucleotides were detected in the cryo-EM structure of the bacterial ESCRT-III Vipp1, PspA and CHMP1B/IST1 filaments were suggested to associated with DNA from the inner side of the tube (Gupta et al, 2021; Junglas et al, 2024; Talledge et al, 2018). Second, the archaeal CDVA protein which is part of the ESCRT homologous archaeal CDV system was reported

to bind DNA (Moriscot et al, 2011). While the exact role of DNA in ESCRT-III filaments assembly and its physiological relevance is yet to be determined, the accumulating evidence supports an interplay between oligonucleotides and the ESCRT-III machinery.

Overall, the tubular Loki ESCRT-III filaments characterized here, share more similar characteristics to bacterial PspA and Vipp1 and human CHMP1B/IST1 filaments than eukaryotic CHMP2/CHMP3 filaments. Filaments of both Asgard ESCRT-III and human CHMP1B, were shown to assemble into helical polymers that exhibit a constricted, more uniform diameter in the presence of an additional ESCRT-III (CHMP1-3 and IST1, respectively); both filaments were reported to bind DNA in vitro; and both deformed membranes via binding to the inner side of the filamentous tube. Consistently, we recently showed that both Loki CHMP4-7 and human CHMP1B associate with chromatin when expressed in mammalian cells (Nachmias et al, 2023). We therefore suggest that Asgard ESCRT-IIIs are functionally more similar to CHMP1B in the eukaryotic ESCRT-III system. The findings that the overall structure of the Loki ESCRT-III helical tube resemble that of the bacterial PspA while the intermolecular interactions between the ESCRT-III helixes are more similar to the ones found in CHMP1B, support the notion that the Asgard ESCRT system is an intermediate between the prokaryotic and eukaryotic systems.

The biological function of Asgard ESCRTs still has to be defined. Our data show that Loki ESCRTs can deform phospholipid-based membranes, while recent characterization of Loki isolates indicate that they obtain a typical archaeal membrane composition, comprising of isoprenoids which are fundamentally different from phospholipids (Imachi et al, 2020). The presence of membrane blebs in Loki isolates suggest a role for ESCRTs in vesicle release. However, release of vesicles requires the canonical topology of budding away from the cytosol, described for ESCRT. Although the here observed binding topology inside the tube suggests the remodeling to be reversed, dynamic membrane remodeling mechanisms can also be envisaged to generate the budding away from the cytosol (Junglas et al, 2021). In addition, whether the membrane remodeling topology observed here represents the topology of ESCRTs in their native environment is to be determined. Future work, investigating ESCRT function in recent Loki isolates and using archaeal-like model membranes is needed to clarify these issues. Regardless of the exact biological function and membrane remodeling topology, our findings that the Asgard ESCRT-III complex bind and remodels membranes and associates with DNA, alongside with its evolutionary conservation makes it a potential candidate for driving core cellular functions that were needed during the emergence

of eukaryotic cells. For example, one possible scenario is that the ESCRT complex was involved in nuclear membrane formation during eukaryogenesis, by bringing together membranes and DNA and facilitating DNA encapsulation by membranes. In this respect, mammalian ESCRT-III proteins have been shown to mediate sealing of the nuclear membrane post cell division and to regulate the attachment of heterochromatin to the newly formed nuclear envelope (Olmos et al, 2015; Pieper et al, 2020; Vietri et al, 2015). Moreover, we recently reported that Loki CHMP4-7 is targeted to the nucleus when overexpressed in eukaryotic cells (Nachmias et al, 2023). Finally, the membrane binding experiments performed here were done using eukaryotic-like phospholipids, which are fundamentally different from archaeal lipids but may resemble the lipid composition of the newly formed nuclear membrane. Therefore, the ESCRT-III system encoded by Asgard archaea could have contributed to the membrane remodeling that occurred during the formation of the phospholipids-based nuclear membrane.

## Methods

### Reagents and tools table

| Reagent/resource | Reference or source | Identifier or catalog number |
|---|---|---|
| **Experimental models** | | |
| **Recombinant DNA** | | |
| | Provided by Eyal Gur | Plasmid psh21 |
| | KKK42122.1 | Obtained from NCBI |
| | KKK44605.1 | Obtained from NCBI |
| **Antibodies** | | |
| | Anti Loki CHMP1-3 | Custom made using the CHMP1-3 protein purified in this study |
| | Anti Loki CHMP4-7 | Custom made using the CHMP1-3 protein purified in this study |
| **Oligonucleotides and other sequence-based reagents** | | |
| ssDNA oligo 40b | ATCCACCTGTACATCAACTC GCCCGGCGGCTCGATCAGCG | |
| ssDNA oligo 20b | GACCCGTTTAGAGGCCCCAA | |
| ssDNA oligo 80b | GTCTGGTGCCACGCGGTAGT GGTGGTATCGAA GGTAGGCAGGAGAATCTGTACTTTCA GGGCGCTAGCCATATGTCATCG | |
| ssDNA oligo 40b no GC | AATAAATTATTTAAATAAAT ATAAATTAAATAAATTATAA | |
| ssDNA oligo poly A 40b | AAAAAAAAAAAAAAAAAAAA AAAAAAAAAAAAAAAAAAAA | |
| ssDNA oligo 160b | ATCCACCTGTACATCAACTC GCCCGGCGGCTCGATCAGCGAT CCACCTGTACATCAACTCGCCCG GCGGCTCGATCAGCGATCCAC CTGTACATCAACTCGCCCGGCG GCTCGATCAGCGATCCACCTG TACATCAACTCGCCCGGCGGCTCGAT CAGCG | |
| ssDNA oligo 200b | ATCCACCTGTACATCAACTCGC CCGGCGGCTCGATCAGCGATCCACCTG TACATCAACTCGCCCGGCGGCTCGATCAG CGATCCACCTGTACATCAACTCGCC CGGCGGCTCGATCAGCGATCCACCTG TACATCAACTCGCCCGGCGGCTCGATCA GCGATCCACCTGTACATCAACTCGCCCG GCGGCTCGATCAGCG | |
| **Chemicals, enzymes, and other reagents** | | |
| | Amylose resin | E8021; NEB, Frankfurt, Germany |
| | Protease inhibitor cocktail | cOmplete ultra-tablets, EDTA-free, Roche |
| | DNase I | 10104159001; Roche Diagnosis GmbH, Manheim, Germany |
| | HisPur™ Ni-NTA Resin | 88222; Thermo Fisher Scientific, Weltham, MA |
| | DOPC | Avanti Polar Lipids Inc.850375 |
| | DOPS | Avanti Polar Lipids Inc.840035 |
| | Uranyl acetate 2% solution | SPI CAS# 6159-44-0 |
| **Software** | | |
| | GraphPad Prism version 9.00 | La Jolla, CA, USA |
| | Velox software version 3.5 | Thermo Fisher Scientific, Inc |
| | BioRender | |
| | imod | University of Colorado |
| | ImageJ | |
| **Other** | | |

### Cloning

Open reading frames (ORFs) from Loki MAG sp. GC14_75, encoding for CHMP1-3 and CHMP4-7 genes were obtained from NCBI (accession numbers KKK42122.1, KKK44605.1, respectively) were amplified by PCR and sub-cloned into pSH21 vector subsequently to the TEV cleavage tag that followed an N terminal polyhistidine and MBP protein tags (a kind gift from Eyal Gur, BGU, Israel) (Elharar et al, 2014). CHMP1-3 ΔC mutant (1-163) was generated by introducing a stop codon after AAs 163 and subcloning the PCR product into the pSH21 vector. All plasmids were verified by sequencing.

### Protein expression and purification

pSH21 plasmids containing Loki CHMP1-3 or CHMP4-7 were expressed Escherichia coli BL21 strain and incubated in LB 50 μg/mL Ampicillin at 37 °C until reaching O.D. of 0.6. Then, induction with 0.5 mM IPTG was performed and the cultures were grown at 30 °C for 3–4 h. Cells were then resuspended and lysed by sonication in buffer A (25 mM HEPES pH = 8, 500 mM NaCl, 10% glycerol, 15 mM 2-mercaptoethanol, 10 mM imidazole) supplemented with protease inhibitor cocktail (cOmplete ultra-tablets, EDTA-free, Roche) and 500 μg/ml DNase I (10104159001; Roche Diagnosis GmbH, Manheim, Germany), and the supernatant was clarified by centrifugation ($25,000 \times g$, 40 min, 4 °C).

For CHMP4-7 purification, the supernatant was subjected to HisPur™ Ni-NTA Resin (88222; Thermo Fisher Scientific, Waltham, MA) column and eluted in buffer B (buffer A supplemented with 50 mM imidazole). Protein tags were then removed using a TEV protease (1 mg/10 mg protein) in buffer C (50 mM Tris-HCL pH = 7.4, 10% glycerol, 500 mM NaCl, 15 mM 2-mercaptoethanol). Finally, a second Ni-NTA step was performed using buffer C, and the flow-through was collected.

For CHMP1-3 purification, the supernatant was subjected to Ni-NTA column and eluted in buffer B. Samples were then subjected to amylose resin (E8021; NEB, Frankfurt, Germany) in buffer D (20 mM Tris pH = 7.4, 500 mM NaCl, 10 mM β-Mercaptoethanol, 10% Glycerol) and eluted in buffer D supplemented with 10 mM maltose.

Protein tags were then removed using a TEV protease (1 mg/10 mg protein) in buffer C (50 mM Tris-HCL pH = 7.4, 10% glycerol, 500 mM NaCl, 15 mM 2-mercaptoethanol). Finally, a second Ni-NTA step was performed using buffer C and the flow-through was collected. Purified proteins were verified my mass spectrometry.

For polymerization, proteins were incubated in buffer E (25 mM Tris pH = 7.6, 50 mM KCl) for 5 h at RT at the indicated conditions. For DNA experiments, buffer E was supplemented with ATCCACCTGTACAT-CAACTCGCCCGGCGGCTCGATCAGCG (40 bases probe, 10 μM), ss or ds at the indicted concentrations. DNase I (10104159001; Roche Diagnosis GmbH, Manheim, Germany) was added at 20 mg/ml concentrations when indicated. Homo polymerization of CHMP1-3 was examined at a range of pH (5.0–8.8), and different salt concentrations (0–500 mM NaCl). Polymerization could not be observed under any of these conditions (Appendix Fig. S1).

## Antibodies

Antibodies for Loki CHMP1-3 and Loki CHMP4-7 were raised in chicken using injection of the full length purifies proteins. Antibodies specificities are shown in Appendix Fig. S6A.

## Preparation of small unilamellar vesicles (SUVs)

Chloroform lipid solutions (DOPC or DOPS) were purchased from Avanti Polar Lipids Inc. (Albaster, AL, USA catalog number 850375, 840035). Lipid mixtures of DOPC/DOPS 1:1 molar ratio and pure DOPC were prepared at a total concentration of 1 mg/ml in Tris 20 mM, KCl 50 mM, pH 7.8 were prepared. Each lipid suspension was extruded 20 times via a 100 nm polycarbonate membrane using a mini-extruder (Avanti Polar Lipids, Alabaster, AL, USA). In the final stage, the vesicles were mixed with the Asgard Loki ESCRT-III filaments. All experiments were done with 8 μM ESCRT-III and 124 μM vesicles in a buffer of Tris 25 mM, KCl 50 mM, at pH 7.6. DNase was added to the indicated conditions at 20 mg/ml concentrations for one hour post vesicle-ESCRT-III incubation.

## Negative-stain grid preparation and imaging

The negative-stain samples for TEM analysis were prepared in the following way: 300 mesh copper grids (Ted Pella, Prod No. 01813-F) were glow-discharged to enhance the hydrophilicity of their surface. Next, 2.5 μL of the sample was applied on to the grid and the excess liquid was blotted with filter paper after 1 min. The grid was dried in air for 1 min, followed by applying 5 μL of uranyl acetate 2% solution (SPI CAS# 6159-44-0) for negative staining to increase the sample contrast. Next, the grid was blotted once more to remove the excess uranyl acetate. Finally, the grid was dried in air before insertion into the microscope. Imaging of the samples was performed with Thermo Fisher Scientific Talos F200C transmission electron microscope operating at 200 kV. The images were taken with Ceta 16 M CMOS camera.

## Cryo-electron microscopy

### Sample preparation

In all, 3 μL of proteins/proteins-phospholipids mix samples were gently deposited on glow-discharged Quantifoil R 1.2/1.3 holey carbon grids (Quantifoil Micro Tools GmbH, Germany). Samples were manually blotted for four seconds at room temperature and vitrified by rapidly plunging into liquid ethane using a home-built plunging apparatus. The frozen samples were stored in liquid nitrogen until imaging.

### Micrograph acquisition

For an initial dataset, samples were loaded under cryogenic conditions and imaged in low dose mode on either a FEI Tecnai F30 Polara microscope (FEI, Eindhoven) operated at 300 kV or a Glacios (Thermo Fisher Scientific) operated at 200 kV. On the Polara, micrographs were loaded under cryogenic conditions and imaged in low dose mode on a FEI Tecnai F30 Polara microscope (FEI, Eindhoven) operated at 300 kV. Micrographs were collected using SerialEM, at a calibrated pixel size of 1.1 Å by a K2 Summit direct electron detector fitted behind an energy filter (Gatan Quantum GIF) set to ±10 eV around zero-loss peak with the total electron dose of 80 ē/Å$^2$. Each dose-fractionated movie had 50 frames, micrographs sums were aligned in SerialEM. On the Glacios, micrographs were collected using Tomo (Single micrograph) at a calibrated pixel size of 0.89 Å by a Falcon 4i direct electron detector fitted behind a Selectris X energy filter set to ±5 eV around zero-loss peak with a total electron dose of 30 e$^-$/Å$^2$. The final dataset used for was collected on 300 kV Titan Krios G4 (Thermo Fisher Scientific) electron microscope equipped with a Biocontinuum K3 (Gatan) detector operated by EPU (Thermo Fisher Scientific) (see details in Table 1). Dose-fractionated movie frames (30 frames per movie) were acquired at a nominal defocus of 1.5–3 μm with a calibrated pixel size of 1.36 Å and a total dose of 30 e$^-$/Å$^2$.

### Helical reconstruction

For the initial dataset, movie frames were gain-corrected, dose-weighted, and aligned using cryoSPARC Live (Punjani et al, 2017). Initial 2D classes were produced using the auto picker implemented in cryoSPARC Live. The following image processing steps were performed using cryoSPARC. The classes with most visible details were used as templates for the filament tracer (50 Å spacing, 50 Å low-pass filter) The resulting filament segments were extracted with 1000 px box size (Fourier cropped to 300 px) and subjected to multiple rounds of 2D classification. The remaining segments were sorted by rod diameter and 2D class averages of the most abundant diameter for initial symmetry guesses in PyHI (Zhang, 2022). Initial symmetry estimates were validated by helical refinement in cryoSPARC and selection of the helical symmetry parameter yielding a reconstruction with typical ESCRT-III features and the best resolution. For the main dataset, movie frames were gain-corrected, dose-weighted, and aligned using cryoSPARC Live (Punjani et al, 2017). Initial 2D classes were produced using the auto picker implemented in cryoSPARC Live. The following image processing steps were performed using cryoSPARC. The classes were separated by filament type (PspA-like rods and curved filaments). Classes with most visible detail were used as templates for the template picker (50 Å spacing). The resulting curved filament segments were extracted with 400 px box size (Fourier cropped to 200 px) and subjected to multiple rounds of 2D classification. The remaining segments were sorted by curvature. Only the least curved segments were kept. No regular helical symmetry could be identified for these filaments, and initial

**Table 1.   Data collection, image processing, and model refinement.**

| | |
|---|---|
| Movies | 12,751 |
| Magnification | 63 kx |
| Voltage (kV) | 300 |
| Total dose (e⁻/ Å²) | 30 |
| Defocus range (μm) | 1.5–3 |
| Physical pixel size (Å) | 1.36 |
| Detector | Gatan K3 |
| Symmetry imposed | 1.05 Å rise 173.9° rotation C1 |
| Final no. of segments (ASUs) | 228,688 (11,205,712) |
| Global map resolution (Å, FSC = 0.143) | 3.55 |
| Local map resolution range (Å, FSC = 0.5) | 3.0–6.0 |
| Initial model used (PDB code) | AF2 prediction |
| **Model refinement** | |
| Model resolution | 3.5 |
| CC mask | 0.76 |
| CC box | 0.29 |
| CC peaks | −0.19 |
| CC volume | 0.78 |
| CC ligands | − |
| Map sharpening B-factor (Å²) | −76 |
| **Model composition** | |
| Nonhydrogen atoms | 1251 |
| Protein | 1251 |
| **RMSDs** | |
| Bond lengths (Å) | 0.007 |
| Bond angles (°) | 0.909 |
| **Validation** | |
| MolProbity score | 1.34 |
| Clashscore | 3.57 |
| Rotamer outliers (%) | 0.74 |
| **Ramachandran plot** | |
| Favored (%) | 96.79 |
| Allowed (%) | 3.21 |
| Disallowed (%) | 0.0 |
| **Deposition IDs** | |
| EMDB | 50583 |
| PDB | 9FN1 |

attempts with ab initio reconstruction did not yield consistent maps. The PspA-like rods were extracted with 800 px box size (Fourier cropped to 200 px) and subjected to multiple rounds of 2D classification. The remaining segments were sorted by diameter. The previously determined helical symmetry was used for an initial reconstruction of the most abundant rods (after re-extraction with 600 px box size, Fourier cropped to 300 px). The corresponding segments were classified by heterogeneous refinement and followed by 3D classifications using the initial helical reconstructions as templates. The resulting helical reconstructions were subjected to multiple rounds of helical refinement including the symmetry search option. For the final polishing, the segments were re-extracted at 600 px without Fourier cropping and subjected to helical refinement including non-uniform refinement. Higher-order aberrations were corrected using global and local CTF-refinement. The local resolution distribution and local filtering for the resulting map was performed using cryoSPARC (as shown in Fig. 4E). The resolution of the final reconstruction was determined by Fourier shell correlation (auto-masked, FSC = 0.143).

### Cryo-EM map interpretation and model building

The handedness of the maps was determined by rigid-body fitting the hairpin of an Alphafold2 (Jumper et al, 2021; Mirdita et al, 2022) prediction of Loki CHMP4-7 into the final map using ChimeraX (Goddard et al, 2018; Pettersen et al, 2021) and flipped accordingly. Predictions of Loki CHMP1–3 and CHMP4-7 monomers were flexibly MDFF fitted to the 3D reconstruction using ISOLDE (Croll, 2018). Only the CHMP4-7 model fitted the secondary structure and large residue pattern present in the map. The CHMP1-3 model was discarded. The CHMP4-7 monomer model was truncated to the parts covered by the map (aa 15–172) and manually refined in ISOLDE. Subsequently, the model was subjected to auto-refinement with *phenix.real_space_refine* (Afonine et al, 2018b). The auto-refined model was checked/adjusted manually in Coot (Emsley et al, 2010). The respective helical symmetry was applied to the model to create assemblies of 60 monomers in ChimeraX and the assembly structure was subjected to another cycle of auto-refinement (with NCS restraints and refinement) using *phenix.real_space_refine* (Afonine et al, 2018b). After the final inspection, the models were validated in *phenix.validation_cryoem* (Afonine et al, 2018a)/Molprobity (Williams et al, 2018).

### Cryo-tomography

Micrographs were acquired using the same setups as of the cryo-EM micrographs. Pixel size at the sample plane on the Polara was 2.3 Å, and 1.9 Å on the Glacios. On the Polara, the camera was operated in counting mode at a dose rate of 6–7 ē/pixels/s, tilt series of 41 exposures from –21° to +60° and back to –60° in 3° intervals and a total dose of ~120 ē /Å² were collected with SerialEM. On the Glacios, the camera was operated in counting mode at a dose rate of 2–3 ē/pixels/sec. Fiducials-free, dose-symmetric tilt series of 61 exposures from –60° to +60° in 2° intervals and a total dose of ~120 ē /Å² were collected with Tomo. Tilt series were aligned, and tomograms were reconstructed using eTomo (IMOD4.11 package) (Mastronarde, 2005). Movies were generated in ImageJ.

## Measurements and statistical analysis

Filament length and diameter measurements were performed in Velox software version 3.5 (Thermo Fisher Scientific, Inc.).

Statistical analysis was performed in GraphPad Prism version 9.00 (La Jolla, CA, USA). Normality was tested using the D'Agostino–Pearson test. For three or more groups, statistical significance was determined by one-way analysis of variance (ANOVA) Kruskal–Wallis test followed by Dunn's post hoc test or by one-way analysis of variance (ANOVA)

followed by Tukey post hoc test, for multiple comparisons. Pearson chi-squared test was used to determine the difference in the distribution of the assembled polymers. Bars in all plots indicate mean values and standard deviation of the mean.

## Data availability

This study includes no data deposited in external repositories.

The source data of this paper are collected in the following database record: biostudies:S-SCDT-10_1038-S44318-024-00346-4.

## Peer review information

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

## Acknowledgements

The authors thank the laboratories of Eyal Gur (BGU) and Eyal Arbely (BGU) for their practical advice on proteins expression and purifications and for kindly sharing their reagents and equipment with us. We also thank Itzik Mizrahi (BGU) and Ori Avinoam (WIS) for constructive feedback on the project. Last, the authors thank all members of the Elia lab for critical feedback throughout the project. This research was funded by the Deutche Forschungsgemeinschaft (DFG, German Research Foundation) project number EL 1199/2-1. The Elia laboratory is funded by the Israeli Science Foundation (ISF) Grant no. 1436/23. ABG is grateful to the Israel Science Foundation for financial support (grant 2101/20). The authors gratefully acknowledge the IKI Institute for Nanoscale Science and Technology, Ben Gurion University of the Negev. The authors are grateful for the generous support from the Guzik Foundation to BGU's Cryo-electron microscopy unit. The authors gratefully acknowledge the electron microscopy training, imaging, and access time granted by the life science EM facility of the Ernst-Ruska Centre at Forschungszentrum Jülich. The authors gratefully acknowledge the computing time granted by the JARA Vergabegremium and provided on the JARA Partition part of the supercomputer JURECA at Forschungszentrum Jülich (Thörnig, 2021).

## Author contributions

**Nataly Melnikov**: Data curation; Methodology. **Benedikt Junglas**: Data curation; Formal analysis; Visualization; Methodology. **Gal Halbi**: Data curation; Methodology. **Dikla Nachmias**: Conceptualization; Data curation; Writing—review and editing. **Erez Zerbib**: Data curation. **Noam Gueta**: Data curation. **Alexander Upcher**: Data curation; Visualization. **Ran Zalk**: Data curation; Visualization. **Carsten Sachse**: Conceptualization; Supervision; Writing—review and editing. **Anne Bernheim-Groswasser**: Funding acquisition; Methodology. **Natalie Elia**: Conceptualization; Funding acquisition; Investigation; Writing—original draft; Project administration; Writing—review and editing.

Source data underlying figure panels in this paper may have individual authorship assigned. Where available, figure panel/source data authorship is listed in the following database record: biostudies:S-SCDT-10_1038-S44318-024-00346-4.

## Disclosure and competing interests statement

The authors declare no competing interests.

