## [Peer Review File · The EMBO Journal]

The Asgard archaeal ESCRT-III system forms helical filaments and remodels eukaryotic-like membranes

Nataly Melnikov, Benedikt Junglas, Gal Halbi, Dikla Nachmias, Erez Zerbib, Noam Gueta, Alexander Upcher, Ran Zalk, Carsten Sachse, Anne Bernheim-Groswasser, and Natalie Elia

Corresponding author(s): Natalie Elia (eliana@post.bgu.ac.il) , Carsten Sachse (c.sachse@fz-juelich.de)

Review Timeline:

Submission Date:	8th Jul 24
Editorial Decision:	13th Aug 24
Revision Received:	14th Oct 24
Editorial Decision:	11th Nov 24
Revision Received:	24th Nov 24
Accepted:	27th Nov 24

Editor: Yehu Moran

Transaction Report:

Dear Dr. Elia,

Thank you for submitting your manuscript for consideration by the EMBO Journal. It has now been seen by three referees whose comments are shown below.

Given the referees' positive recommendations, I would like to invite you to submit a revised version of the manuscript, addressing the comments of all three reviewers. I should add that it is EMBO Journal policy to allow only a single round of revision, and acceptance of your manuscript will therefore depend on the completeness of your responses in this revised version.

I would suggest that after going over the comments and discussing them with your co-authors we will meet for a video call to discuss your perspective on the comments and the potential plan for revisions. We usually suggest this when the Referees ask for additional experiments and we would like to set with the authors realistic expectations for the revision.

Thank you for the opportunity to consider your work for publication. I look forward to your revision.

Yours sincerely,

Yehu Moran
Editor
The EMBO Journal

- a Reagents and Tools Table as part of the Methods section, which can be downloaded from our author guidelines

(<https://www.embopress.org/page/journal/14602075/authorguide#structuredmethods>)

We realize that it is difficult to revise to a specific deadline. In the interest of protecting the conceptual advance provided by the work, we recommend a revision within 3 months (11th Nov 2024). Please discuss the revision progress ahead of this time with the editor if you require more time to complete the revisions. Use the link below to submit your revision:

Referee #1:

Elia and colleagues describe the polymerization of two the Asgard ESCRT-III proteins, CHMP1-3 and CHMP4-7. They show that CHMP4-7 assembles into helical tubular structures alone and in the presence of CHMP1-3. Assembly is somewhat controlled by ssDNA in vitro. The authors present the cryoEM structure of CHMP4-7/CHMP1-3 tubes, which reveal only the presence of CHMP4-7 in an open ESCRT-III conformation. Polymerization resembles both bacterial and eukaryotic ESCRT-III assemblies that can bind membrane inside the protein tube. Experiments with LUVs confirms that membrane can enter inside the protein tube, which is consistent with structural elements such as helix0 positioned on the inside to interact with membranes and the C-terminus on the outside and thus likely accessible to VPS4 for disassembly. The role of ssDNA is less clear and no DNA seems to be present in the structure. Furthermore, the absence of CHMP1-3 in the structure poses more questions.

In summary, this is an important manuscript that provides a first view on the structural conservation of ESCRT-III in Asgard. Because Asgard express only two ESCRT-III proteins, understanding this simplified system will greatly help to understand the function of ESCRT-III polymers and will inform on the evolutionary relationship of ESCRT-III.

The following concerns should be addressed prior to publication:

The manuscript insists a lot on the role of ssDNA oligonucleotides. However, it seems that this could be an in vitro artefact that somehow helps nucleation of the polymer. Although there is accumulating evidence that ESCRT-III may merit its previous name "CHromatin Modifying Protein -CHMP", but this would be probably limited to dsDNA. Hence, I suggest to tone down the role of ssDNA and its speculative physiological function.

CHMP1-3 is described as changing the polymer structure of CHMP4-7. However, it is not present/visible in the cryoEM map/structure. The authors should provide other evidence that CHMP1-3 is indeed an integral part of the polymer either by SDS-PAGE analysis or mass spec or another approach.

Is the density inside the tubes continuous? What's the estimated distance to the positively charged side chains? Can the authors provide Mass spec evidence that it could be DNA?

Did the authors test assembly with longer DNA - several hundreds of base pairs?

The structure indicates that CHMP4-7 binds positively curved membrane. There are only two ESCRT-III proteins; how can this be reconciled with the role of ESCRT-III in cell division, which requires the interaction with negatively curved membranes.

Minor points:

Line 300: "deform negatively charged membrane via the interior of the tube" ... This statement is misleading; the polymer does not interact/coat negatively charged membrane. It only shows that membrane can enter or gets drawn inside in vitro!

Line 392: "CDVA was reported to bind DNA "- please add citation

Line 724: Azad et al is published in NSMB.

Referee #2:

The manuscript "The Asgard archaeal ESCRT-III system forms helical filaments and remodels eukaryotic membranes, shedding light on the emergence of eukaryogenesis" reports cryo-EM structure of the ESCRT III filaments represented by a protein from a Lokiarchaeum (Loki) of Asgard superphylum. It has been found that the filaments share some features with both bacterial and eukaryotic ESCRT III filaments. Authors demonstrated that Asgard ESCRT III filaments can interact and deform eukaryotic-like membrane vesicles in vitro. Several additional experiments were performed to show how different conditions affect this interaction. In particular it has been shown that oligonucleotides facilitate the assembly of ESCRT-III filaments but inhibit membrane remodeling. Experimental demonstration that Asgard ESCRT-III can remodel eukaryotic membrane is important for understanding a transition from archaea to eukaryotes despite the fundamental differences in their lipid composition. Overall, I believe that this is an interesting study and the manuscript is clearly written. I have rather minor comments and suggestions.

1. I think the paper would be more impactful if authors could compare interactions of Asgard ESCRT III with SUV composed of archaea-like synthetic phospholipids.
2. I suggest to modify the title: "shedding light on eukaryogenesis" or "shedding light on the origin of eukaryotes". Although I think that this is an overstatement. The fact that Asgard ESCRT remodels eukaryotic membranes tell us a little about involvement of ESCRT systems in the origin of endomembrane systems of eukaryotes. It likely can remodel bacterial membranes too suggesting that this is a non-specific function and on the other hand presence of ESCRT in Asgard did not lead to emergence of eukaryote-like endomembrane systems in those organisms.
3. The trees on both graphical abstract and Fig. 1 are misleading because they show "Archaea" for one of the branches while other archaea are shown elsewhere. You can say "other archaea" instead or draw the tree without this "loose" branch and indicate "archaea" for entire clade containing eukaryotes and all archaeal branches you want to show (2D tree).
4. Generally, I think that graphical abstract should better reflect the content of the paper, which is about the ESCRT filament structure and remodeling of SUV composed of eukaryotic phospholipids.
5. In discussion mention the most obvious function of ESCRT - ubiquitinated protein sorting. The link of ESCRT genes with ubiquitin pathway genes is undeniable and described in several papers.
6. Line 68. evolutionary theories -> evolutionary hypothesis or evolutionary analysis; This is not a theory yet.
7. Line 71. Cite doi: 10.1038/s41586-021-03494-3 and doi: 10.1038/s41586-023-06186-2
8. Line 79. Cite doi: 10.1128/mbio.00335-2
9. Line 90. superfamilies -> complexes
10. Line 107. Cite doi: 10.1128/mbio.00335-24
11. Line 139. "models suggests that eukaryotes integrate from" -> "model suggests that eukaryotes derived from"
12. Line 142. Explain what is Snf7 domain; if needed, show it on a figure.
13. Line 279. Delete comma.
14. Lines 391-392. Include references.

Referee #3:

The manuscript from Melnikov et al. demonstrates that ESCRT homologs identified in Asgard archaea, prokaryotes with eukaryotic signature proteins, are able to assemble into helical filaments, suggesting functional similarity to the eukaryotic ESCRT machinery. The authors demonstrate a variety of filament morphologies obtained in the presence of varying partners, present a 3.6 Å reconstruction of CHMP4-7 filaments from cryoEM data and provide data suggesting their interaction with small unilamellar vesicles (SUVs)

This is an intriguing study that in principle would be suitable for publication in the EMBO J. As it stands, however, there are a number of issues that suggest the work is at a rather preliminary stage:

- Much of the motivation / discussion is devoted to the Asgard ESCRT system as yielding clues to eukaryogenesis (a kind of "missing link"). As the proteins are from extant organisms, this argument appears to me to be spurious. The proteins under study probably have a function of their own within the organisms, one that is not discussed here at all
- Although the filaments take on varying morphologies in the presence of CHMP1-3 and ssDNA and the reconstruction was from such mixtures, the latter entities are not detected in the reconstruction. A discussion of why this might be the case (the authors have previously suggested [ref. 26] that CHMP4-7 might correspond to human CHMP1B and CHMP1-3 to IST1, which can each form filaments as well as co-filaments [ref. 30]) is missing. For the human system, it has previously been reported [ref. 18] that helical order could be improved with a truncated IST1 construct.
- Considering the range of filament diameters, it would be helpful to see the power spectra of the 2D class averages in Figs. 1D 4B. Are the authors certain that these represent single filament species?
- In Fig. 3B (control), the lower left corner would appear to show a multi-layered filament reminiscent of CHMP1B-IST1 co-filaments - could the authors comment on this?
- The role of nucleic acids in both in vitro filament assembly as well as physiologically in vivo is unclear to me. Although this has also been observed in the eukaryotic system [ref. 26], the latter study corresponds to a preprint (posted in 2018) that has not been certified by peer review. The question arises as to whether this represents a physiological interaction (which would indeed be very interesting), or whether this is due to polyanion binding. The ssDNA sequence used in this study was ATCCACCTGTACATCAACTCGCCCGGCGGCTCGATCAGCG - was there a rationale behind the choice of sequence? The previous study [26] showed a featureless "tube" of density - maybe e.g. polyA/polyG etc. could be used to identify any possible specificity?

- I am not sure that I find the SUV interaction data entirely convincing. If feasible, confocal light microscopy of lipid-labelled GUVs with the variously labelled protein components would go a long way towards demonstrating filament formation from vesicular material. It seems that the vesicles shown in Fig. 6A (middle panel) appear to have a (protein?) coat - is this the case?
- Considering that ESCRT proteins are sensitive to lipid composition, it would be helpful to discuss what (if anything) is known about Asgard membrane compositions and how this might affect the Asgard ESCRT system.

Department of Life Sciences
Prof. Natalie Elia
October 14, 2024

Department of Life Sciences
Ben Gurion University of the Negev
Referee #1:

Elia and colleagues describe the polymerization of two the Asgard ESCRT-III proteins, CHMP1-3 and CHMP4-7. They show that CHMP4-7 assembles into helical tubular structures alone and in the presence of CHMP1-3. Assembly is somewhat controlled by ssDNA in vitro. The authors present the cryoEM structure of CHMP4-7/CHMP1-3 tubes, which reveal only the presence of CHMP4-7 in an open ESCRT-III conformation. Polymerization resembles both bacterial and eukaryotic ESCRT-III assemblies that can bind membrane inside the protein tube. Experiments with LUVs confirms that membrane can enter inside the protein tube, which is consistent with structural elements such as helix0 positioned on the inside to interact with membranes and the C-terminus on the outside and thus likely accessible to VPS4 for disassembly. The role of ssDNA is less clear and no DNA seems to be present in the structure. Furthermore, the absence of CHMP1-3 in the structure poses more questions.

In summary, this is an important manuscript that provides a first view on the structural conservation of ESCRT-III in Asgard. Because Asgard express only two ESCRT-III proteins, understanding this simplified system will greatly help to understand the function of ESCRT-III polymers and will inform on the evolutionary relationship of ESCRT-III.

The following concerns should be addressed prior to publication:

The manuscript insists a lot on the role of ssDNA oligonucleotides. However, it seems that this could be an in vitro artefact that somehow helps nucleation of the polymer. Although there is accumulating evidence that ESCRT-III may merit its previous name "CHromatin Modifying Protein -CHMP", but this would be probably limited to dsDNA. Hence, I suggest to tone down the role of ssDNA and its speculative physiological function.

We agree with the reviewer that, at this point, we are unable to indicate whether the DNA associated phenotype is physiologically relevant. We had specifically indicated this in the discussion (see revised paragraph on DNA, page 14 last paragraph and specifically lines 447-448 on page 15). Also, we have omitted two paragraphs from the discussion in which the role of DNA was discussed.

CHMP1-3 is described as changing the polymer structure of CHMP4-7. However, it is not present/visible in the cryoEM map/structure. The authors should provide other evidence that CHMP1-3 is indeed an integral part of the polymer either by SDS-PAGE analysis or mass spec or another approach.

Department of Life Sciences
Prof. Natalie Elia
October 14, 2024

We thank the reviewer for this comment. We have recently generated specific antibodies for CHMP1-3 and CHMP4-7 using the purified proteins. We employed these antibodies to demonstrate that CHMP1-3 reside in the filament fraction using both denaturative and native gels followed by western blot analysis (Fig. S6A-B). Additionally, we added a paragraph to the discussion summarizing our accumulating data on CHMP1-3 and their potential implications (page 13 last paragraph, starting line 402).

Is the density inside the tubes continuous? What's the estimated distance to the positively charged side chains? Can the authors provide Mass spec evidence that it could be DNA?

As presented in the greyscale density section of former Fig. S4C (now Fig. S5D), the weak cylindrical density present in the tube lumen is continuous and it does not follow the imposed helical symmetry. The density has a distance of 25 Å to the closest positively charged residues R16. We now added a statement and show the radial density profile (Fig. S5D) to clarify this item. Following our biochemical approach for visualizing the proteins in the Loki ESCRT-III filament fraction, we have also demonstrated the presence of ssDNA in the filament fraction using agarose gels (Fig. S6C).

Did the authors test assembly with longer DNA - several hundreds of base pairs?

To better understand the basis of DNA binding we have analyzed oligos comprising different length and compositions (see new supplementary figure - Fig. S4). This analysis revealed that 40-80 long sequences containing GC increases Loki ESCRT-III polymerization while longer sequences (160, 200 bases) and sequences lacking GC do not.

The structure indicates that CHMP4-7 binds positively curved membrane. There are only two ESCRT-III proteins; how can this be reconciled with the role of ESCRT-III in cell division, which requires the interaction with negatively curved membranes.

We agree with the reviewer that the data presented here for Loki ESCRT-III and elsewhere for PspA, Vipp1 and IST1/CHMP1B raise mechanistic questions regarding ESCRT-III membrane binding in different cellular processes. It is possible that observed phenotypes result from the in vitro set-up, and we now clearly indicate that in the discussion (page 14 starting line 418 and page 16 lines 477-481). That said, they may very well represent the physiological membrane binding properties of these ESCRT homologs and may point to additional ESCRT functions / alternative mechanisms. To simplify things and avoid confusion, we omitted the paragraph in the discussion that describes potential role for ESCRTs in cell division.

Minor points:

Department of Life Sciences
Prof. Natalie Elia
October 14, 2024

Line 300: "deform negatively charged membrane via the interior of the tube" ... This statement is misleading; the polymer does not interact/coat negatively charged membrane. It only shows that membrane can enter or gets drawn inside in vitro!

Line 392: "CDVA was reported to bind DNA "- please add citation

Line 724: Azad et al is published in NSMB.

Thank you for these comments. This is now fixed in the revised version

Referee #2:

The manuscript "The Asgard archaeal ESCRT-III system forms helical filaments and remodels eukaryotic membranes, shedding light on the emergence of eukaryogenesis" reports cryo-EM structure of the ESCRT III filaments represented by a protein from a Lokiarchaeum (Loki) of Asgard superphylum. It has been found that the filaments share some features with both bacterial and eukaryotic ESCRT III filaments. Authors demonstrated that Asgard ESCRT III filaments can interact and deform eukaryotic-like membrane vesicles in vitro. Several additional experiments were performed to show how different conditions affect this interaction. In particular it has been shown that oligonucleotides facilitate the assembly of ESCRT-III filaments but inhibit membrane remodeling. Experimental demonstration that Asgard ESCRT-III can remodel eukaryotic membrane is important for understanding a transition from archaea to eukaryotes despite the fundamental differences in their lipid composition. Overall, I believe that this is an interesting study and the manuscript is clearly written. I have rather minor comments and suggestions.

1. I think the paper would be more impactful if authors could compare interactions of Asgard ESCRT III with SUV composed of archaea-like synthetic phospholipids.

We agree with the reviewer that using archaeal-like model membranes would advance our functional knowledge on Asgard ESCRTs. However, while protocols for generating phospholipid-based SUVs are readily available, generating archaeal based model membranes is far more challenging – synthetic versions of archaeal lipids are lacking, and generating SUVs from such lipids requires the establishment of new methodologies. We are currently collaborating with experts in the field of membrane biophysics and hopefully will be able to address this question in the future. We now refer to this point in the discussion (page 16 lines 468-474).

2. I suggest to modify the title: "shedding light on eukaryogenesis" or "shedding light on the origin of eukaryotes". Although I think that this is an overstatement. The fact that Asgard ESCRT

Department of Life Sciences
Prof. Natalie Elia
October 14, 2024

remodels eukaryotic membranes tell us a little about involvement of ESCRT systems in the origin of endomembrane systems of eukaryotes. It likely can remodel bacterial membranes too suggesting that this is a non-specific function and on the other hand presence of ESCRT in Asgard did not lead to emergence of eukaryote-like endomembrane systems in those organisms.

We have removed the part on eukaryogenesis from the title.

3. The trees on both graphical abstract and Fig. 1 are misleading because they show "Archaea" for one of the branches while other archaea are shown elsewhere. You can say "other archaea" instead or draw the tree without this "loose" branch and indicate "archaea" for entire clade containing eukaryotes and all archaeal branches you want to show (2D tree).

We have revised Fig. 1A to address this point.

4. Generally, I think that graphical abstract should better reflect the content of the paper, which is about the ESCRT filament structure and remodeling of SUV composed of eukaryotic phospholipids.

We modified the graphical abstract to include the structural data.

5. In discussion mention the most obvious function of ESCRT - ubiquitinated protein sorting. The link of ESCRT genes with ubiquitin pathway genes is undeniable and described in several papers.

We agree that there is a clear link between the ubiquitin sorting system and ESCRTs that can be traced to Asgard archaea and that this may be the key for understanding ESCRT function in this archaeal domain. We now explicitly mention that in the introduction (page 4 lines 113-116). That said, as our work was focused on the ESCRT-III complex we feel that we did not make a contribution to this angle and therefore decided not to focus on this point further.

6. Line 68. evolutionary theories -> evolutionary hypothesis or evolutionary analysis; This is not a theory yet.

7. Line 71. Cite doi: 10.1038/s41586-021-03494-3 and doi: 10.1038/s41586-023-06186-2

8. Line 79. Cite doi: 10.1128/mbio.00335-2

9. Line 90. superfamilies -> complexes

10. Line 107. Cite doi: 10.1128/mbio.00335-24

11. Line 139. "models suggests that eukaryotes integrate from" -> "model suggests that eukaryotes derived from"

12. Line 142. Explain what is Snf7 domain; if needed, show it on a figure.

13. Line 279. Delete comma.

Department of Life Sciences
Prof. Natalie Elia
October 14, 2024

14. Lines 391-392. Include references.

We incorporated all suggestions, except for #8 because we were unable to locate the citation. We will be happy to incorporate it if more details are provided. Regarding the comment #12 on snf7, we decided to omit this from the text for simplicity.

Referee #3:

The manuscript from Melnikov et al. demonstrates that ESCRT homologs identified in Asgard archaea, prokaryotes with eukaryotic signature proteins, are able to assemble into helical filaments, suggesting functional similarity to the eukaryotic ESCRT machinery. The authors demonstrate a variety of filament morphologies obtained in the presence of varying partners, present a 3.6 Å reconstruction of CHMP4-7 filaments from cryoEM data and provide data suggesting their interaction with small unilamellar vesicles (SUVs). This is an intriguing study that in principle would be suitable for publication in the EMBO J. As it stands, however, there are a number of issues that suggest the work is at a rather preliminary stage:

- Much of the motivation / discussion is devoted to the Asgard ESCRT system as yielding clues to eukaryogenesis (a kind of "missing link"). As the proteins are from extant organisms, this argument appears to me to be spurious. The proteins under study probably have a function of their own within the organisms, one that is not discussed here at all

We fully agree with the reviewer that the function of the proteins within the organism should also be discussed. We now added information on recent Loki Asgard isolates to the introduction (page 5 paragraph starting line 129) and discussed the potential role of ESCRTs based on these publications (page 16 starting line 468).

- Although the filaments take on varying morphologies in the presence of CHMP1-3 and ssDNA and the reconstruction was from such mixtures, the latter entities are not detected in the reconstruction. A discussion of why this might be the case (the authors have previously suggested [ref. 26] that CHMP4-7 might correspond to human CHMP1B and CHMP1-3 to IST1, which can each form filaments as well as co-filaments [ref. 30]) is missing. For the human system, it has previously been reported [ref. 18] that helical order could be improved with a truncated IST1 construct.

We thank the reviewer for this comment. To gain more insights into the role of CHMP1-3 we have generated a C' truncated CHMP1-3 mutant. Similar to full-length CHMP1-3 the mutant did not polymerize on its own. Additionally, adding the CHMP1-3 mutant to CHMP4-7 inhibited polymerization. These findings support direct interactions between CHMP1-3 and CHMP4-7 and indicate that removing the C terminus of CHMP1-3 does not facilitate polymerization (see data

Department of Life Sciences
Prof. Natalie Elia
October 14, 2024

in Fig. S2B, S3C). We further performed biochemical analysis of the filament fraction using custom made antibodies for Loki ESCRT-IIIs that were recently generated in our lab. These data further support association of CHMP1-3 with the filament (Fig. S6). Finally, we tried to determine the cryo-EM structure of the CHMP4-7 homopolymer alone. However, we found that the obtained CHMP4-7 preparations were too heterogeneous and variable to enable high resolution reconstructions, further supporting a role for CHMP1-3 in structuring the ESCRT-III tube. Overall, we find these results unexpected, and we are keen on understanding the function of CHMP1-3 in these two components ESCRT-III system. We added a paragraph to the discussion summarizing our accumulating data on CHMP1-3 and their potential implications (paragraph starting at page 13 line 402).

- Considering the range of filament diameters, it would be helpful to see the power spectra of the 2D class averages in Figs. 1D 4B. Are the authors certain that these represent single filament species?

We now include the power spectra of the 2D class averages with increasing diameters 320, 340, 360, 380, 400 and 420 Å in Fig. S5C. Based our analysis, the helical tubes in the sample represent the same basic filament architecture, while they simply increase the number of units per turn.

- In Fig. 3B (control), the lower left corner would appear to show a multi-layered filament reminiscent of CHMP1B-IST1 co-filaments - could the authors comment on this?

Although we saw this phenotype occasionally, it does not represent the majority of the helical tubes in the sample. We therefore replaced the image in Fig 3B with a more representative one.

- The role of nucleic acids in both in vitro filament assembly as well as physiologically in vivo is unclear to me. Although this has also been observed in the eukaryotic system [ref. 26], the latter study corresponds to a preprint (posted in 2018) that has not been certified by peer review. The question arises as to whether this represents a physiological interaction (which would indeed be very interesting), or whether this is due to polyanion binding. The ssDNA sequence used in this study was ATCCACCTGTACATCAACTCGCCCGCGGCTCGATCAGCG - was there a rationale behind the choice of sequence? The previous study [26] showed a featureless "tube" of density - maybe e.g. polyA/polyG etc. could be used to identify any possible specificity?

We thank the reviewer for this constructive comment. We have now tested the effect of different oligos on Loki ESCRT-III polymerization. We found that poly-A oligos and oligos lacking the GC nucleotides do not induce polymerization, indicating that the effect is not simply polyanion binding and that there is at least some sequence specificity associated with this effect (Fig. S4A). That said, we fully agree that the physiological role of DNA binding is yet to be established and

Department of Life Sciences
Prof. Natalie Elia
October 14, 2024

we now clearly state this in the discussion (page 15 line 447). We further removed parts in the discussion involving DNA to avoid overspeculations.

- I am not sure that I find the SUV interaction data entirely convincing. If feasible, confocal light microscopy of lipid-labelled GUVs with the variously labelled protein components would go a long way towards demonstrating filament formation from vesicular material. It seems that the vesicles shown in Fig. 6A (middle panel) appear to have a (protein?) coat - is this the case?

The EM-based assay we used was previously shown to describe membrane remodeling by bacterial ESCRTs (Junglas, 2021). The fluorescence-based GUV assay, which provides dynamic information, is perhaps more commonly used, yet it does not provide the spatial resolution obtained by cryo-EM. Although we think it is important to characterize the dynamics of ESCRT binding on GUVs, this will require establishing a new experimental setup in our laboratory and is outside the scope of the current work. We do realize that distiguishing between proteins and lipids is somewhat more challenging in cryo-EM tomography and we have, therefore, replaced the previous images in the main fig. with an improved tomogram and added artificial coloring of the membrane and proteins to improve visualization (Fig. 6D and sup movie 3).

As for the possibility of having a protein coat on the vesicle in Fig. 6A – this data was obtained by negative staining and we could not find evidence for this in cryo-EM data – which exhibit higher resolution and sample preservation. Hence, we do not think proteins coat the vesicle in the docking stage.

- Considering that ESCRT proteins are sensitive to lipid composition, it would be helpful to discuss what (if anything) is known about Asgard membrane compositions and how this might affect the Asgard ESCRT system.

We have added this information to the discussion (page 16 starting line 468).

Dear Prof. Elia,

Thank you for submitting your manuscript for consideration by the EMBO Journal. It has now been seen by the three original referees whose comments are enclosed. As you will see, all three referees express interest in your manuscript and are broadly in favour of publication, pending satisfactory minor revision by one of them.

Given the referees' positive recommendations, I would like to invite you to submit a revised version of the manuscript, addressing the comments by Referee #3.

Additionally, please address the specific comments provided below by our editorial team.

We generally allow three months as standard revision time. As a matter of policy, competing manuscripts published during this period will not negatively impact on our assessment of the conceptual advance presented by your study. Yet, I hope that since the requested revision is not including any additional experiments you would be able to address all comments in a much shorter time.

Thank you for the opportunity to consider your work for publication. I look forward to your revision.

Yours sincerely,

Yehu Moran
Academic Editor
The EMBO Journal

Specific comments for your manuscript by editorial team:

- *Keywords: missing, please add.
- *DATA AVAILABILITY SECTION: missing, please add.
- *AC: remove from manuscript text
- *Disclosure & competing interest statement: please rename
- *DATASET EV LEGENDS: Please correct the movie title to "Movie EV1" - EV3. Please zip each movie file with a short legend in simple readme format.
- *APPENDIX 1 FILE WITH ToC: Please rename the PDF with supplementary figures "Appendix" and add a table of contents, including page numbers. Please correct the nomenclature to "Appendix Figure S1" etc., and also update the callouts in the manuscript text accordingly.
- *SOURCE DATA: missing: Hannah Sonntag sent an email with her request on 22 August. Please make sure to address this request.
- *SYNOPSIS IMAGE: provided in the manuscript; please upload it as a separate file in png or jpg format, sized 550 pixels wide x 300 - 600 pixels high
- *SYNOPSIS TEXT: not provided. Please provide.

Notes:

- Please remove Table 1 from the supplementary information PDF and add to the manuscript text, in editable format, after the main figure legends.

- DAS:

Please add a data availability statement at the end of Methods. Please include a direct link to the deposited data, as well as a reviewer access code, if applicable.

- Figure legends:

1. Please define the annotated p values ****/***/* as well as provide the exact p-values for the same in the legend of figure 2b; 3d; as appropriate.
2. Please indicate the statistical test used for data analysis in the legend of figure 2b.
3. Please note that n=2 in figure 3c.
4. Although 'n' is provided, please describe the nature of entity for 'n' in the legends of figures 2b; 3d.

5. Please note that the error bars are not defined in the legends of figures 2b; 3d.

General instructions for preparing your revised manuscript:

- a point-by-point response to the referees' comments, with a detailed description of the changes made (as a word file).

- a word file of the manuscript text.

- individual production quality figure files (one file per figure)

- a complete author checklist, which you can download from our author guidelines

(<https://www.embopress.org/page/journal/14602075/authorguide>).

- Expanded View files (replacing Supplementary Information)

- a Reagents and Tools Table as part of the Methods section, which can be downloaded from our author guidelines

(<https://www.embopress.org/page/journal/14602075/authorguide#structuredmethods>)

We realize that it is difficult to revise to a specific deadline. In the interest of protecting the conceptual advance provided by the work, we recommend a revision within 3 months (9th Feb 2025). Please discuss the revision progress ahead of this time with the editor if you require more time to complete the revisions. Use the link below to submit your revision:

Referee #1:

The authors have well responded to my previous concerns and I support publication of the manuscript in its current form.

Referee #2:

I am satisfied with the revised manuscript and have no further suggestions

Referee #3:

The revised manuscript from Melnikov et al. has been considerably improved, taking into accounts the previous comments of the reviewers. Nevertheless, there remain a number of issues that should be addressed prior to acceptance:

• A glance at the revised evolutionary tree (Figure 1A) makes clear that the archaeal Asgard clade is not an ancestor of eukaryotes, just that they share a common (late) ancestor. As no function(s) has/have yet been assigned to these Asgard proteins, it is not possible to interpret any similarities and differences in structure to extant prokaryotic and eukaryotic proteins. As such, the following statements remain unsupported and should be deleted or modified:

o Line 48 (abstract): "substantiating a role for ESCRTs in eukaryogenesis"

o Lines 126/127: ", and could have been involved in membrane remodelling processes that occurred during eukaryogenesis"

o Lines 158-159: "Current phylogenetic models suggest that eukaryotes derived from the archaeal Asgard clade (Fig. 1A)."

o Lines 300-304: "Secondly, in the bacterial ESCRT-III the linker between $\alpha 3$ and $\alpha 4$ is longer and contains more helix breaking residues (i.e., G156, G157 and G159 in Vipp1). Therefore, the Loki CHMP4-7 assembly more closely resembles the bacterial

configuration with the characteristic spike, while the Loki CHMP4-7 monomer adopts a conformation closer to previously determined eukaryotic structures including the $\alpha 1/2-\alpha 5$ interaction motif."

o Lines 365-367: "we demonstrate that the Asgard ESCRT-III could, in-principle, contribute to the membrane remodeling processes that occurred during eukaryogenesis and gave rise to the complex phospholipids-based endomembrane system of eukaryotes."

o Lines 388-391: "For eukaryotes and Loki archaea, we detected a number of residue omissions in helices $\alpha 2+\alpha 3$ and the linker between $\alpha 3$ and $\alpha 4$, whose associated properties appear to be dispensable in the context of ESCRT evolution and may be responsible for functional diversification of different isoforms."

o Lines 461-467: "We therefore suggest that Asgard ESCRT-IIIs are functionally more similar to CHMP1B in the eukaryotic ESCRT-III system. The findings that the overall structure of the Loki ESCRT-III helical tube resemble that of the bacterial PspA while the intermolecular interactions between the ESCRT-III helices are more similar to the ones found in CHMP1B, support the notion that the Asgard ESCRT system is an intermediate between the prokaryotic and eukaryotic systems."

o Lines 484-499: "alongside with its evolutionary conservation makes it an exceptional candidate for driving core cellular functions that were needed during the emergence of eukaryotic cells. For example, one possible scenario is that the ESCRT complex was involved in nuclear membrane formation during eukaryogenesis, by bringing together membranes and DNA and facilitating DNA encapsulation by membranes."

• The significance of nucleic acid binding continues to escape me. Although the authors provide additional data using other sequences and oligonucleotide lengths, I do not see this as sufficient to ascribe a functional role to ssDNA binding. Indeed, the authors' earlier publication from Nachmias et al. (2020) describes binding of dsDNA (which to my mind would be more consistent with chromatin binding). The authors should explain this discrepancy. The presence of ssDNA in the filament fraction (lines 265-267, Figure S6) does not of course demonstrate its presence in the filaments themselves. Personally, I could envisage ssDNA oligonucleotide binding as mimicking phospholipid binding on a membrane, which would also be consistent with the positive charge of $\alpha 0$ (line 279). Were any experiments made to explore Loki CHMP4-7 and/or Loki CHMP1-3 tube formation in the presence of acidic SUVs (which is known to enhance helix formation of IST1/CHMP1B, McCullough et al., 2015) without nucleic acids (in contrast to pre-assembled filaments)? The following statements should be modified or deleted:

o Lines 215-216: "Recent work from our laboratory showed that purified Loki CHMP4-7 binds short DNA oligonucleotides (Nachmias et al., 2022)."

o Lines 238-239: "further supporting a role for DNA in ESCRT-III helical tube assembly (Fig. 3E and Fig. S3E)."

o Lines 279-281: "Noteworthy, $\alpha 0$ is positively charged and could possibly interact with DNA in the lumen of the rods to form the observed cylindrical density."

o Lines 362-363: "We further provide data to support a role for DNA in the assembly of Asgard ESCRT-III filaments."

o Line 433: "Our results demonstrate a role for DNA in Loki ESCRT-III polymerization."

o Lines 448-451: "While the exact role of DNA in ESCRT-III filaments assembly and its physiological relevance is yet to be determined, the accumulating evidence supports the notion that oligonucleotides binding is another common property of the ESCRT-III machinery."

• The idea that Loki CHMP1-3 might possess a scaffolding-like function (supported by the lack of polymerisation upon including a C-terminally truncated variant) is intriguing and attractive. If the protein is indeed contributing to tube nucleation, one might expect that CHMP1-3 molar ratios significantly lower than 1:6 CHMP4-7 could seed polymerization, which is an experiment worth trying. As mentioned above for the biochemical analyses of ssDNA, antibody binding in the filament fraction (lines 265-267, Figure S6) is not the same as the protein being present in the filament; this should be made clear to the reader.

• The density for helix $\alpha 5$ (Figure 5B) does not appear to be sufficiently resolved to be able to place side chains unambiguously. A comparison with an e.g. AlphaFold3 model might be illuminating, as well as a description of the interface formed between $\alpha 5$ and $(\alpha 1/\alpha 2)_{j+4}$, and how this compares with other ESCRT polymers.

• The structure-based sequence alignment shown in Figure S5H reveals a number of insertions and deletions within helices, which runs counter to (my) intuition. Corresponding structural overlays of the individual helices would therefore be helpful.

• The relevance of the section on cultivated Asgard Loki cultivations (lines 129-136) to the present work is not clear to me. Are membrane protrusions and blebs unique amongst Archaea? Or to organisms containing ESCRT proteins?

• As far as I can see, the "canonical ESCRT-III fold" is made of 5 α -helices and not 6 (lines 386-387)

• Replace "memubiqbrane" with "membrane" (line 473)

• In Figure 3, the yellow lines obscure the tubes; please swap these with their corresponding (original) images shown in Figure S3.

Department of Life Sciences
Prof. Natalie Elia
November 24, 2024

Dear Prof. Moran,

We are pleased to submit our revised version for the manuscript entitled “The Asgard archaeal ESCRT-III system forms helical filaments and remodels eukaryotic-like membranes”. In the revised version, we fully addressed the comments raised by the editorial team and have implemented minor changes to the text following the suggestions of reviewer 3. Also, to address the comments of reviewer 3 we introduced a new panel to Sup Fig. 3 showing binding of ssDNA to CHMP 4-7 using a gel-shift assay. Finally, we noticed a typo in first name of one of the co-authors, which we amended in the revised version.

We hope you will find the revised version suitable for publication in *EMBO journal*.

Below please find a point-by-point response to the reviewers' comments.

Sincerely,

Natalie Elia, PhD
Department of Life Sciences
Ben Gurion University of the Negev

Department of Life Sciences
Prof. Natalie Elia
November 24, 2024

Referee #1:

The authors have well responded to my previous concerns and I support publication of the manuscript in its current form.

Referee #2:

I am satisfied with the revised manuscript and have no further suggestions

Referee #3:

The revised manuscript from Melnikov et al. has been considerably improved, taking into accounts the previous comments of the reviewers. Nevertheless, there remain a number of issues that should be addressed prior to acceptance:

- A glance at the revised evolutionary tree (Figure 1A) makes clear that the archaeal Asgard clade is not an ancestor of eukaryotes, just that they share a common (late) ancestor. As no function(s) has/have yet been assigned to these Asgard proteins, it is not possible to interpret any similarities and differences in structure to extant prokaryotic and eukaryotic proteins. As such, the following statements remain unsupported and should be deleted or modified:

It is true that the relation between Asgard archaea and eukaryotes is still a hypothesis that is based on bioinformatics analysis. Because of that, we think it is important to relate experimental data to the model, and to highlight common features between Asgard and eukaryotic proteins. A similar approach has been taken by others as well (e.g. PMID 35697693, 32747565).

o Line 48 (abstract): "substantiating a role for ESCRTs in eukaryogenesis" we have deleted this part from the abstract.

Department of Life Sciences
Prof. Natalie Elia
November 24, 2024

o Lines 126/127: ", and could have been involved in membrane remodelling processes that occurred during eukaryogenesis"

We deleted this part.

o Lines 158-159: "Current phylogenetic models suggest that eukaryotes derived from the archaeal Asgard clade (Fig. 1A)." We revised the text to: "Current phylogenetic models suggest that eukaryotes branched from a lineage closely related to the archaeal Asgard clade" (Fig. 1A) And added a relevant citation.

o Lines 300-304: "Secondly, in the bacterial ESCRT-IIIs the linker between $\alpha 3$ and $\alpha 4$ is longer and contains more helix breaking residues (i.e., G156, G157 and G159 in Vipp1). Therefore, the Loki CHMP4-7 assembly more closely resembles the bacterial configuration with the characteristic spike, while the Loki CHMP4-7 monomer adopts a conformation closer to previously determined eukaryotic structures including the $\alpha 1/2$ - $\alpha 5$ interaction motif." –

This is a straightforward comparison between the structures. We therefore decided to keep it as is.

o Lines 365-367: "we demonstrate that the Asgard ESCRT-III could, in-principle, contribute to the membrane remodeling processes that occurred during eukaryogenesis and gave rise to the complex phospholipids-based endomembrane system of eukaryotes."

We realize that there is a lot to be done for this to be confirmed yet given the current suggested relations between Asgard archaea and eukaryotes we think it is important to highlight this aspect in the discussion.

o Lines 388-391: "For eukaryotes and Loki archaea, we detected a number of residue omissions in helices $\alpha 2 + \alpha 3$ and the linker between $\alpha 3$ and $\alpha 4$, whose associated properties appear to be dispensable in the context of ESCRT evolution and may be responsible for functional diversification of different isoforms."

This sentence is part of the discussion and is directly related to the structural data obtained here and their comparison to other high-resolution structures published previously for eukaryotic ESCRTs. The term "evolution" refers to changes in the protein structure that occurred throughout evolution.

Department of Life Sciences
Prof. Natalie Elia
November 24, 2024

o Lines 461-467: "We therefore suggest that Asgard ESCRT-IIIs are functionally more similar to CHMP1B in the eukaryotic ESCRT-III system. The findings that the overall structure of the Loki ESCRT-III helical tube resemble that of the bacterial PspA while the intermolecular interactions between the ESCRT-III helices are more similar to the ones found in CHMP1B, support the notion that the Asgard ESCRT system is an intermediate between the prokaryotic and eukaryotic systems." The first part of the sentence is simply comparing our new structure of Loki-ESCRT-III to previously published ESCRT-III structures. The last part is referring to this comparison in relation to the current model and is written in a suggestive manner. As this is part of the discussion, we see no reason to modify it.

o Lines 484-499: "alongside with its evolutionary conservation makes it an exceptional candidate for driving core cellular functions that were needed during the emergence of eukaryotic cells. For example, one possible scenario is that the ESCRT complex was involved in nuclear membrane formation during eukaryogenesis, by bringing together membranes and DNA and facilitating DNA encapsulation by membranes."

We replaced "exceptional" with "potential".

- The significance of nucleic acid binding continues to escape me. Although the authors provide additional data using other sequences and oligonucleotide lengths, I do not see this as sufficient to ascribe a functional role to ssDNA binding. Indeed, the authors' earlier publication from Nachmias et al. (2020) describes binding of dsDNA (which to my mind would be more consistent with chromatin binding). The authors should explain this discrepancy. The presence of ssDNA in the filament fraction (lines 265-267, Figure S6) does not of course demonstrate its presence in the filaments themselves. Personally, I could envisage ssDNA oligonucleotide binding as mimicking phospholipid binding on a membrane, which would also be consistent with the positive charge of $\alpha 0$ (line 279). Were any experiments made to explore Loki CHMP4-7 and/or Loki CHMP1-3 tube formation in the presence of acidic SUVs (which is known to enhance helix formation of IST1/CHMP1B, McCullough et al., 2015) without nucleic acids (in contrast to pre-assembled filaments)? The following statements should be modified or deleted:

Department of Life Sciences
Prof. Natalie Elia
November 24, 2024

We agree that the role of DNA in physiological context should be further substantiated, and this is clearly stated in the text (line 451). That said, we now show that the sequence of oligonucleotides significantly influences their effect on tube formation. Therefore, the possibility that the observed phenotype solely stems from unspecific binding of the negatively charged phosphate groups to positively charged residues in the protein is unlikely. As for DNA binding shown in our previous publication - it is true that the previous experiment was done using ds oligoes. We have repeated this experiment using ssDNA and got similar results. We added this data to Fig. S3 (Fig. S3C). Finally, regarding the suggested assay of adding the membrane during filaments assembly – indeed this can complement the findings reported here. However, it requires adaptation of the self-assembly protocol we have developed. We are now working toward modifying our polymerization assay so that we could address this in future studies.

o Lines 215-216: "Recent work from our laboratory showed that purified Loki CHMP4-7 binds short DNA oligonucleotides (Nachmias et al., 2022)."

This statement refers to our previous publication and is backed up by experimental data.

o Lines 238-239: "further supporting a role for DNA in ESCRT-III helical tube assembly (Fig. 3E and Fig. S3E)."

We modified this sentence to "further supporting a role for DNA in ESCRT-III helical tube assembly, in-vitro"

o Lines 279-281: "Noteworthy, $\alpha 0$ is positively charged and could possibly interact with DNA in the lumen of the rods to form the observed cylindrical density."

We deleted the words "DNA in" from the sentence to avoid overspeculations.

o Lines 362-363: "We further provide data to support a role for DNA in the assembly of Asgard ESCRT-III filaments."

This statement is part of the discussion. We have toned down the statement by replacing the word "support" with "suggest".

o Line 433: "Our results demonstrate a role for DNA in Loki ESCRT-III polymerization."

We modified this sentence to: "Our results demonstrate that DNA can facilitate Loki ESCRT-III polymerization"

Department of Life Sciences
Prof. Natalie Elia
November 24, 2024

o Lines 448-451: "While the exact role of DNA in ESCRT-III filaments assembly and its physiological relevance is yet to be determined, the accumulating evidence supports the notion that oligonucleotides binding is another common property of the ESCRT-III machinery." We have modified the last part of the sentence to: "the accumulating evidence supports an interplay between oligonucleotides and the ESCRT-III machinery".

- The idea that Loki CHMP1-3 might possess a scaffolding-like function (supported by the lack of polymerisation upon including a C-terminally truncated variant) is intriguing and attractive. If the protein is indeed contributing to tube nucleation, one might expect that CHMP1-3 molar ratios significantly lower than 1:6 CHMP4-7 could seed polymerization, which is an experiment worth trying. As mentioned above for the biochemical analyses of ssDNA, antibody binding in the filament fraction (lines 265-267, Figure S6) is not the same as the protein being present in the filament; this should be made clear to the reader.

This is clearly stated in lines 410-412.

- The density for helix $\alpha 5$ (Figure 5B) does not appear to be sufficiently resolved to be able to place side chains unambiguously. A comparison with an e.g. AlphaFold3 model might be illuminating, as well as a description of the interface formed between $\alpha 5_j$ and $(\alpha 1/\alpha 2)_{j+4}$, and how this compares with other ESCRT polymers.

We agree with the reviewer's comment that the density of helix \$\alpha 5\$ is between 4.0 and 5.5 local Å resolution and does not allow reliable model building and refinement. We stated in the methods (page 20) that an AlphaFold2 model was placed into the density and followed by model refinement. As requested by the referee, we present an overlay of an AlphaFold3 model (red) and our structure (cyan). The helix backbone overlay very well while some of the side chains vary in their conformation. These differences can be expected at this local lower resolution. Due to the limited information on helix \$\alpha 5\$ in the cryo-EM structure, we refrain from further structural comparisons with other ESCRT-III proteins.

Department of Life Sciences
Prof. Natalie Elia
November 24, 2024

- The structure-based sequence alignment shown in Figure S5H reveals a number of insertions and deletions within helices, which runs counter to (my) intuition. Corresponding structural overlays of the individual helices would therefore be helpful.

Response: we appreciate the concern about the “counterintuitive” deletions and insertions of the reviewer on the presented sequence alignment. As stated in the caption, we used T-coffee to generate the alignment of sequences, which does not use structural information. In this context of low evolutionary conservation, insertions and deletions on this magnitude are not surprising, which is, of course, also reflected in the structures. Therefore, more structural overlays are not adding to the understanding of the evolutionary emergence.

- The relevance of the section on cultivated Asgard Loki cultivations (lines 129-136) to the present work is not clear to me. Are membrane protrusions and blebs unique amongst Archaea? Or to organisms containing ESCRT proteins?

Yes, these are typical ESCRT functions. We have modified the text to clarify this point. Interestingly, both species are characterized with elongated membrane protrusions and membrane blebs, which are aligned with typical ESCRT functions.

- As far as I can see, the "canonical ESCRT-III fold" is made of 5 α -helices and not 6 (lines 386-387).

Department of Life Sciences
Prof. Natalie Elia
November 24, 2024

The original literature on the structure of ESCRT-III proteins refer to 6 helices. However, because the conserved fold obtained from recent cryo-EM structures appear to include five helices we decided to modify this sentences as suggested.

- Replace "memubiqbrane" with "membrane" (line 473). Thank you for noticing that.
- In Figure 3, the yellow lines obscure the tubes; please swap these with their corresponding (original) images shown in Figure S3.

We find it hard for people who are not from the field to visualize the phenotype without the highlighted yellow lines. All the raw images are provided in the sup. figures so that more experienced readers can evaluate the data for themselves.

Dear Prof. Elia,

I am pleased to inform you that your manuscript has been accepted for publication in the EMBO Journal.

Yours sincerely,

Yehu Moran
Academic Editor
The EMBO Journal
